# Conservative Offline Policy Adaptation in Multi-Agent Games

**Chengjie Wu**[1], **Pingzhong Tang**[12], **Jun Yang**[3], **Yujing Hu**[4],
**Tangjie Lv**[4], **Changjie Fan**[4], **Chongjie Zhang**[5]
[1]Institute for Interdisciplinary Information Sciences, Tsinghua University
[2]Turingsense
[3]Department of Automation, Tsinghua University
[4]Fuxi AI Lab, NetEase
[5]Department of Computer Science & Engineering, Washington University in St. Louis
wucj19@mails.tsinghua.edu.cn
{kenshin,yangjun603}@tsinghua.edu.cn
{huyujing,hzlvtangjie,fanchangjie}@corp.netease.com
chongjie@wustl.edu

## Abstract

Prior research on policy adaptation in multi-agent games has often relied on online interaction with the target agent in training, which can be expensive and impractical in real-world scenarios. Inspired by recent progress in offline reinforcement learning, this paper studies offline policy adaptation, which aims to utilize the target agent's behavior data to exploit its weakness or enable effective cooperation. We investigate its distinct challenges of distributional shift and risk-free deviation, and propose a novel learning objective, conservative offline adaptation, that optimizes the worst-case performance against any dataset consistent proxy models. We propose an efficient algorithm called Constrained Self-Play (CSP) that incorporates dataset information into regularized policy learning. We prove that CSP learns a near-optimal risk-free offline adaptation policy upon convergence. Empirical results demonstrate that CSP outperforms non-conservative baselines in various environments, including Maze, predator-prey, MuJoCo, and Google Football.

## 1 Introduction

Reinforcement learning (RL) has shown promise in learning collaborative or adversarial policies in multi-agent games, including cooperative multi-agent reinforcement learning [49, 6, 34, 5, 14, 40], and learning in adversarial environments such as Texas hold'em and MOBA games [3, 45, 46, 50]. Research also recognizes the importance of adapting policy to different players in multi-agent games [10, 26, 51]. For instance, in zero-sum games, although an approximate Nash equilibrium strategy optimizes worst-case performance, it can be overly conservative when competing with opponents of limited rationality and loses opportunities to exploit [27, 17, 24]. This problem is known as opponent exploitation [17, 23, 41, 1, 28]. In cooperative games, the agent may be required to cooperate with other competent players that have diverse behaviors [15, 11], instead of just a specific group of teammates. The problem is investigated in recent literature such as ad-hoc cooperation [11] and zero-shot coordination [15, 26, 51, 47]. In this paper, **policy adaptation** refers to the general ability to learn to collaborate with or exploit other participants (called target agents) in multi-agent games.

This paper focuses on a specific problem of **offline policy adaptation**. In this setting, interaction with the target agent is not available in training. Instead, we leverage the target agent's behavior dataset and maintain access to the game environment during training. The problem definition is illustrated

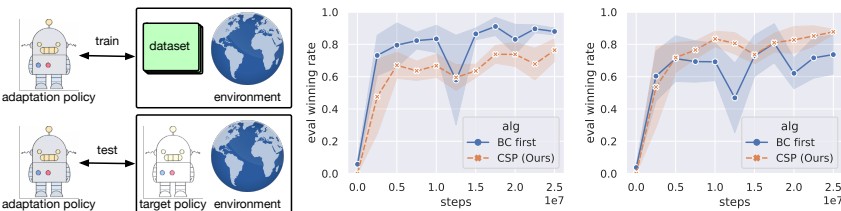

Figure 1: **Left.** Problem illustration. **Mid and right.** Winning rates in Google Football 3vs1 (attacker). The x-axis is the training steps. We run evaluations against **(mid)** proxy model in training and **(right)** real target agent. Shaded areas represent standard error. The non-conservative method quickly overfits the proxy model. Our method CSP outperforms baseline in testing. Please refer to Appendix G.7 for more explanations.

in Figure 1 (left). This setting is suitable for many practical scenarios where offline information about the target agent is provided. It lies between the fully transparent case, where the target agent's policy is available, and the fully invisible zero-shot adaptation case. For instance, in competitions such as Go, soccer, and MOBA games, playing with the target opponent is not an option until the actual game commences. However, professional human players have the intelligence to analyze their opponents' and teammates' strategic characteristics by studying past records, enhance their skills, and make adaptations accordingly before the competition. Moreover, training AIs with humans in applications that involve human-AI interaction is exorbitant. Additionally, it can be unsafe and poses ethical risks when humans interact with an inadequately trained AI in certain circumstances.

Although offline policy adaptation models a wide range of applications, it has not been adequately investigated both theoretically and empirically in previous literature. For instance, some opponent exploitation research assumes direct interaction in training for explicit opponent model estimation [17, 24] or policy gradient calculation [2, 10, 12]. However, this assumption is strong because it allows queries of the target agent's policy for any input. Other works study exploiting totally unknown opponents [7] or collaborating with completely unseen teammates [26, 51, 47] in a zero-shot manner. These methods do not take advantage of possible prior information about the target agent, and their performances heavily depend on the construction of a diverse and representative policy population constructed for training, which itself is a challenging problem [31, 25].

In this paper, we identify and formalize the distinct challenges in offline policy adaptation, and show that previous opponent exploitation methods, which disregard these challenges, are inadequate for effectively addressing the problem. Firstly, offline policy adaptation inherits the *distributional shift* [22] challenge from traditional offline RL. A straightforward approach to address offline policy adaptation is by initially creating an estimation of the target agent (e.g., utilizing behavior cloning) from data and subsequently learning an adaptation policy based on the proxy model. We name this method *BC-First*. However, since the dataset usually will not expose all aspects of the real policy of the target agent due to limited dataset size, the proxy model can differ arbitrarily from the real target on out-of-distribution (OOD) states. The discrepancy between the proxy in training and the real target in testing causes the distributional shift problem for the adaptation policy. As shown in Figure 1, *BC-First* overfits the dataset, which leads to erroneous generalization on OOD states and results in significant performance degradation in evaluation. Our algorithm outperforms the baseline and improves in both training and testing as the number of training step increases. Secondly, since the adaptation policy has access to the environment in training, unlike standard offline RL, we show that "conservatism" [8, 20, 19] for addressing distributional shift may not be sufficient in this setting. It is possible to benefit from deviation in multi-agent games if the worst-case rewards obtainable outside the dataset exceed those inside the dataset. We coin this challenge as *risk-free deviation*.

To address these challenges, we propose a novel learning objective, **conservative offline adaptation**, which optimizes the worst-case performance when maintaining a conservative stance towards the target agent on the policy level. This objective provides a unified approach to addressing the challenges of distributional shift and risk-free deviation. To optimize this objective, we introduce a simple yet efficient algorithm, **Constrained Self-Play (CSP)**, which simultaneously trains an adaptation policy and a proxy model subject to regularizations. We prove that CSP produces a near-optimal risk-free adaptation policy upon convergence, given that the target agent's policy within the support of the dataset can be well approximated. Empirically, we evaluate the effectiveness of our algorithm in four environments: a didactic maze environment, predator-prey in MPE [39], a competitive two-agent

MuJoCo environment [2, 10] requiring continuous control, and Google Football [21]. Our results show that non-conservative methods confront difficulties regarding offline policy adaptation. The experiment results demonstrate that our algorithm significantly outperforms baselines.

## 2 Related Work

**RL in multi-agent games.** Multi-agent reinforcement learning (MARL) under the centralized training decentralized execution (CTDE) paradigm have shown promising performances in learning coordination behavior for cooperative multi-agent tasks such as SMAC [35] and Google Football [21]. Representative works consist of policy gradient algorithms [6, 48], and value decomposition methods [34, 40]. There also have been extensive research in competitive environments such as Texas hold'em and MOBA games. Some works design theoretically sounded RL algorithms which are guaranteed to converge to approximate NE [3]. Some works propose RL methods based on self-play [45, 46, 50] to train max-min policies which show strong empirical performances.

**Offline RL.** Offline reinforcement learning [22] learns purely from batched data. Representative works take a conservative view of out-of-distribution state-action pairs to mitigate the distributional shift problem. BCQ [8] uses VAE to propose candidate actions that are within the support of the dataset. IQL [19] uses expectile regression to learn Q function to avoid querying out-of-distribution actions. CQL [20] puts regularization on the learning of Q function to penalize out-of-distribution actions.

**Opponent modeling.** Opponent modeling, sometimes also called opponent exploitation, is a broad and extensively studied area [1, 28]. The offline adaptation problem studied in this paper is a largely unstudied yet significant and challenging problem within the broader scope of opponent modeling. Some previous works assume access to the target agent's policy in training, and employ deep reinforcement learning to exploit the opponent in continuous control tasks using policy gradients estimated from direct interaction [2, 10, 12]. Some works, including RNR [17] and SES [24] study safe exploitation given an estimated opponent model. They keep the exploitation policy close to NE in order to minimize its own exploitability. Some papers, like GSCU [7], study zero-shot generalization ability to exploit totally unknown opponents. Additionally, this paper is orthogonal to some other opponent modeling subareas, including designing efficient models for predicting opponent behavior [13, 33], and addressing non-stationary opponents [7]. To our best knowledge, there lacks a thorough investigation into the offline adaptation problem.

Please refer to Appendix B for more discussion.

## 3 Preliminary

For simplicity, we consider a two-player fully observable game $G = (N, S, A, P_M, r, p_0, \gamma)$ [37, 38], where $N$ is a set of two players, $S$ is a set of states, $p_0$ is initial state distribution, and $\gamma$ is the discount factor. We use $A_i$ to denote the available actions for player $i$, and $A = (A_1, A_2)$. The transition probability is denoted by $P_M : S \times A \times S \to \mathbb{R}$. We use $r_i : S \times A \to \mathbb{R}$ to denote the reward function for player $i$, and $r = (r_1, r_2)$. In fully cooperative scenarios, $r_1(s, \boldsymbol{a}) = r_2(s, \boldsymbol{a}) = r(s, \boldsymbol{a})$, which is termed multi-agent MDP (MMDP) [43]. In zero-sum games, $r_1(s, \boldsymbol{a}) + r_2(s, \boldsymbol{a}) = 0$. Player $i$'s policy $\pi_i(s, \cdot)$ is a distribution over action space $A_i$ conditioned on state $s \in S$. The value function $V_i^{(\pi_1, \pi_2)}(s)$ is the expected discounted accumulated reward that player $i$ will achieve under joint policy profile $(\pi_1, \pi_2)$ at state $s$. Similarly, the action value function $Q_i^{(\pi_1, \pi_2)}(s, \boldsymbol{a})$ calculates the expected return when players play the joint action $\boldsymbol{a} = (a_1, a_2)$. Let $d_{\pi_1, \pi_2}(s)$ be the discounted visitation frequency of state $s$ under policy profile $(\pi_1, \pi_2)$: $d_{\pi_1, \pi_2}(s) = \sum_{t=0}^{\infty} \gamma^t P(s_t = s)$. We use $J_i^{(\pi_1, \pi_2)}$ to denote the return for player $i$ under policy profile $(\pi_1, \pi_2)$.

Without loss of generality, assume that we are aimed at learning an adaptation policy for player 1 (P1), while player 2 (P2), called the target agent, is playing a unknown fixed strategy $\pi_B$. In offline policy adaptation, we do not have access to $\pi_B$. Instead, a dataset $D$ of $\pi_B$'s behavior data is given. Formally, $D = F_p(\{\mathcal{T}_i\})$, where $F_p$ denotes extra data processing[1], and $\mathcal{T}_i = \{s_0, a_0, s_1, \dots\}$ records states and corresponding actions of an episode collected by $\pi_B$ playing with rollout policies in

---

[1]We use $F_p$ to represent possible information loss in data collection, e.g., truncating and randomly masking.

$\Pi_c$. We do not make assumptions on the data collecting process, therefore, $\Pi_c$ can contain any polices ranging from a weak random policy to a strong NE policy. Analogous to offline RL and robust RL, the dataset quality varies given different rollout policies, dataset sizes, and processing methods, which results in different problem difficulties. For instance, if the dataset reveals adequate information about the target's behavior under diverse situations, BC-First can be enough. The challenges of offline policy adaptation becomes prominent if the dataset only contains limited knowledge of $\pi_B$.

The purpose of offline adaptation is to learn to adapt to unknown $\pi_B$ given a fixed batch of data $D$. Notably, different from offline RL, the access to the environment itself is still granted in training. If not, such setting can be reduced to standard imitation learning or offline reinforcement learning [44, 29], because the target's behavior can be absorbed into environment dynamics. With the advantage of having access to the environment, this setting encourages conservative exploitation whenever risk-free deviation is possible.

## 4 Conservative Offline Adaptation

To address the problem of offline adaptation, we first examine the extrapolation error in policy evaluation between joint policies $(\pi, \pi_B)$ and $(\pi, \mu)$ in section 4.1. Here, $\pi$ denotes the learning policy, $\pi_B$ represents the true target policy, and $\mu$ denotes any proxy model that estimates $\pi_B$. This extrapolation error poses a risk to non-conservative policy adaptation algorithms, rendering them unsafe. However, traditional offline RL methods that aim to minimize the impact of extrapolation error tend to be excessively conservative, which is suboptimal in this context. To allow for beneficial deviations from the dataset, we introduce a novel objective of conservative offline adaptation in section 4.2. This approach maximizes the worst-case performance against any dataset-consistent proxy (defined below). Finally, we propose a practical Constrained Self-Play (CSP) algorithm in section 4.3 and formally prove that it achieves near-optimal conservative offline adaptation. Please note that all proofs pertaining to this section can be found in Appendix A.

### 4.1 Extrapolation Error in Offline Adaptation

We first analyze the extrapolation error in policy evaluation between the joint policies $(\pi, \pi_B)$ and $(\pi, \mu)$, which results in the overfitting problem shown in Figure 1. For theoretical analysis, we first introduce the dataset consistent policy in Definition 4.1. We will show that this assumption can be relaxed in algorithm design in a later section. Inspired by offline RL, in Proposition 4.2, we find that the extrapolation error in offline policy adaptation can also be decomposed in a Bellman-like recursion [8, 44]. We focus solely on the value function of our learning agent (Player 1), and omit the subscript in the rewards and value functions for simplicity.

**Definition 4.1.** *Dataset consistent policy.* A policy $\mu$ is said to be consistent with $\pi_B$ on dataset $D$ iff. $\mu(s, a) = \pi_B(s, a)$, $\forall (s, a) \in D$. We denote the set of all dataset consistent policies as $\mathcal{C}_D$.

**Proposition 4.2.** *The performance gap of evaluating policy profile $(\pi, \mu)$ and $(\pi, \pi_B)$ at state $s$ is $\varepsilon(s) = V^{\pi, \pi_B}(s) - V^{\pi, \mu}(s)$, which can be decomposed in a Bellman-like recursion:*

$$\varepsilon(s) = \sum_{a_1, a_2, s'} \pi(a_1|s) P_M(s'|s, \boldsymbol{a}) \left( \pi_B(a_2|s) - \mu(a_2|s) \right) \left( r(s, \boldsymbol{a}, s') + \gamma V^{\pi, \mu}(s') \right)$$
$$+ \sum_{a_1, a_2, s'} \pi(a_1|s) \pi_B(a_2|s) P_M(s'|s, \boldsymbol{a}) \gamma \varepsilon(s') \tag{1}$$

**Theorem 4.3.** *We use $\varepsilon = \sum_{s_0} p_0(s_0) \varepsilon(s_0)$ to denote the overall performance gap between policy profile $(\pi, \mu)$ and $(\pi, \pi_B)$. In any 2-player fully observable game, for all reward functions, $\varepsilon = 0$ if and only if $\pi_B(a|s) = \mu(a|s)$, $\forall s, s.t. d_{\pi, \pi_B}(s) > 0$.*

Theorem 4.3 proves that the extrapolation error vanishes if and only if the estimation $\mu$ perfectly matches $\pi_B$ on any state which can be visited by $(\pi, \pi_B)$. If extrapolation error exists, improvement of $\pi$ against proxy $\mu$ in training will not guarantee monotonic improvement in testing when $\pi$ is evaluated with $\pi_B$ (as shown in Figure 1). Because we make no assumptions on $\pi_B$ outside $D$, we cannot bound $\|\mu(a|s) - \pi_B(a|s)\|$ for $s \notin D$. Therefore, it requires that $d_{\pi, \pi_B}(s) = 0, \forall s \notin D$.

**Corollary 4.4.** *Given a dataset consistent policy $\mu \in \mathcal{C}_D$, which is consistent with $\pi_B$ on dataset $D$, the extrapolation error vanishes if $\forall s \notin D$, $d_{\pi, \pi_B}(s) = 0$.*

Corollary 4.4 advocates excessive conservatism, similar to standard offline RL, to exclude any state-action pairs outside the dataset. It requires to constrain both $\pi$ and $\mu$ such that the trajectories of $(\pi, \mu)$ stay within $D$. However, this approach comes at the cost of relinquishing opportunities to exploit environmental access and explore the potential benefit of risk-free deviation from the dataset.

## 4.2 Conservative Offline Adaptation

In order to avoid being overly conservative and allow beneficial deviation from the dataset, we propose a novel learning objective called conservative offline adaptation in Definition 4.5.

**Definition 4.5.** *Conservative offline adaptation (COA).* Given an unknown policy $\pi_B$, and a dataset $D$ of its behavior data, conservative offline adaptation optimizes the worst-case performance against any possible dataset-consistent policy:

$$\max_{\pi} \min_{\mu} J(\pi, \mu), \ s.t. \ \mu \in \mathcal{C}_D. \tag{2}$$

The adaptation policy $\pi^*$ is an optimal risk-free adaptation policy if it is a solution to Objective (2).

To show that COA enables risk-free deviation, we decompose the return into two parts in Definition 4.6: (1) the return that is solely realized within the dataset, and (2) the return that arises from deviation.

**Definition 4.6.** *Within-dataset return & off-dataset return.* The expected return of a joint policy profile $(\pi_1, \pi_2)$ can be decomposed into the sum of within-dataset return $J_D(\pi_1, \pi_2)$ and off-dataset return $J_F(\pi_1, \pi_2)$:

$$
\begin{aligned}
J(\pi_1, \pi_2) &= \mathbb{E}_{\tau \sim (\pi_1, \pi_2)} \sum_t \gamma^t r(s_t, \boldsymbol{a}_t, s_{t+1}) \\
&= \mathbb{E}_{\tau \sim (\pi_1, \pi_2)} \sum_{t \leq t_D^\tau} \gamma^t r(s_t, \boldsymbol{a}_t, s_{t+1}) + \mathbb{E}_{\tau \sim (\pi_1, \pi_2)} \sum_{t > t_D^\tau} \gamma^t r(s_t, \boldsymbol{a}_t, s_{t+1}) \\
&= J_D(\pi_1, \pi_2) + J_F(\pi_1, \pi_2),
\end{aligned} \tag{3}
$$

where $t_D^\tau = \arg\max_{t'} \{\forall t \leq t', s_t \in D\}$ is a random variable indicates the last time step in a trajectory $\tau$ such that the state is still contained in dataset $D$.

For any policy $\pi$, and any dataset consistent policy $\mu \in \mathcal{C}_D$, $J_D(\pi, \mu) = J_D(\pi, \pi_B)$ because $\mu$ and $\pi_B$ behave exactly the same until the first time an OOD state is reached according to the definition. Thus, the objective 2 is equivalent to $\max_\pi J_D(\pi, \pi_B) + \min_\mu J_F(\pi, \mu), \ s.t. \ \mu \in \mathcal{C}_D$. The adaptation policy $\pi$ learns to exploit or cooperate with the target agent's policy exposed by dataset $D$. Meanwhile, the max-min optimization of $\max_\pi \min_\mu J_F(\pi, \mu)$ encourages $\pi$ to explore opportunities outside the dataset while maintaining a conservative stance towards the target agent at the policy level. It provides a unified approach to automatically decide whether to deviate depending on the magnitude of within-dataset return and worst-case off-dataset return. In the extreme case, if $\min_\mu J_F(\pi, \mu)$ is much larger than $J_D(\pi, \pi_B)$, $\pi$ can ignore the dataset completely and try to obtain rewards outside the dataset. Additionally, suppose that $(\pi^*, \mu^*)$ is the solution to COA, then the extrapolation error $\varepsilon = J(\pi^*, \pi_B) - J(\pi^*, \mu^*) = J_F(\pi^*, \pi_B) - J_F(\pi^*, \mu^*) \geq 0$, which suggests that the testing performance will not be worse than training performance.

## 4.3 Constrained Self-Play

Directly optimizing the objective of COA is hard because in deep reinforcement learning, satisfying the dataset consistency assumption regarding the proxy model $\mu$ can be difficult, particularly for stochastic policies with limited data. We relax this assumption and propose a simple yet efficient algorithm, Constrained Self-Play (CSP). We prove that CSP can achieve near-optimal risk-free policy adaptation. We start by showing in Theorem 4.7 that the gap in return is bounded by the KL divergence of two policies of the target agent.

**Theorem 4.7.** *In a two-player game, suppose that $\pi$ is player 1's policy, $\alpha$ and $\mu$ are player 2's policies. Assume that $\max_s D_{\mathrm{KL}}(\alpha(\cdot|s)\|\mu(\cdot|s)) \leq \delta$. Let $R_M$ be the maximum magnitude of return obtainable for player 1 in this game, and let $\gamma$ be the discount factor. We use $J(\pi, \mu)$ and $J(\pi, \alpha)$ to denote the return for player 1 when playing policy profiles $(\pi, \mu)$ and $(\pi, \mu)$ respectively. Then,*

$$J(\pi, \mu) - J(\pi, \alpha) \geq -R_M \sqrt{2\delta} \left(1 + \frac{2\gamma\delta}{(1-\gamma)^2}\right). \tag{4}$$

With this result, we propose to optimize objective 5, which substitutes the hard dataset consistent constraint in objective 2 with KL-divergence.

$$\max_\pi \min_\mu J(\pi, \mu) \ s.t. \ \max_{s \in D} D_{\mathrm{KL}}(\pi_B(\cdot|s)\|\mu(\cdot|s)) \leq \delta \tag{5}$$

We propose the Constrained Self-Play (CSP) algorithm, which optimizes objective 5 through learning both an adaptation policy and a proxy model with self-play and meanwhile minimizing the KL divergence between $\mu$ and $\pi_B$ on the dataset with regularization. The algorithm is shown in Algorithm 1 in Appendix C. CSP utilizes soft BC regularization to minimize the KL-divergence: $\max_\pi \min_\mu \left\{ J(\pi, \mu) + C_{\mathrm{BC}} \cdot \mathbb{E}_{(s,a)\sim D}[-\log \mu(a|s)] \right\}$. It is because BC is equivalent to KL-divergence minimization [18, 9]. By increasing the coefficient $C_{\mathrm{BC}}$, CSP prioritizes KL-divergence minimization. In the max-min optimization, the proxy model is trained adversarially against our adaptation policy by setting the proxy's reward function to be the negative of our agent's reward. We use MAPPO [48] as the basic learning algorithm for players on both sides.

We prove in Theorem 4.8 that CSP approximates the optimal risk-free offline adaptation policy by some constant upon convergence. The gap depends on the maximum KL-divergence between the learned proxy $\mu$ and the target $\pi_B$ on $s \in D$. The gap decreases as $\mu$ becomes a more precise estimation of $\pi_B$ within $D$. As a sanity check, the gap vanishes when $\mu$ perfectly matches $\pi_B$ in $D$.

**Theorem 4.8.** *Let $\pi^*$ be the optimal risk-free offline adaptation policy at the convergence of the optimization of objective 2, and let $\tilde{\pi}$ be the policy at the convergence of objective 5. Then the worst-case adaptation performance of $\tilde{\pi}$ is near-optimal:*

$$\min_{\mu \in \mathcal{C}_D} J(\pi^*, \mu) \geq \min_{\mu \in \mathcal{C}_D} J(\tilde{\pi}, \mu) \geq \min_{\mu \in \mathcal{C}_D} J(\pi^*, \mu) - R_M\sqrt{2\delta}\left(1 + \frac{2\gamma\delta}{(1-\gamma)^2}\right). \tag{6}$$

## 5   Didactic Maze Example

In this section, we use an adversarial grid-world maze game to demonstrate the challenges posed by offline policy adaptation and explain the superiority of our algorithm. Figure 2 displays the maze environment, where Player 1 (P1) and Player 2 (P2) start from the labeled positions and can move in four directions within the maze. The grids marked with numbers are terminal states with indicated rewards. There are five corridors (2 for P1 and 3 for P2), each with different lengths and exponential final rewards[2]. For simplicity, we use the notation P2 $\rightarrow$ 16 to denote P2's policy which going downwards to the state with terminal reward 16. The first agent reaching any terminal state wins the reward, and the opponent receives the negative reward as a penalty. The game ends immediately. We assume that P1 wins if P1 and P2 arrive at terminal states simultaneously. If no one reaches a terminals in 10 steps, they both get a penalty of -500. Both agents have full observation of the game states. Assume that P1 is the opponent, and we want to find exploiting policies for P2.

We consider four baselines: (1) **BC-First** as explained in section 1; (2) **Self-Play (max-min)**: approximating the pure max-min policy of P2 using self-play, which does not exploit; (3) **Dataset-Only**: completely following the dataset; and (4) **Oracle**: training an exploitation policy directly against the real target $\pi_B$. Please refer to Appendix D for detailed learning curves in this environment.

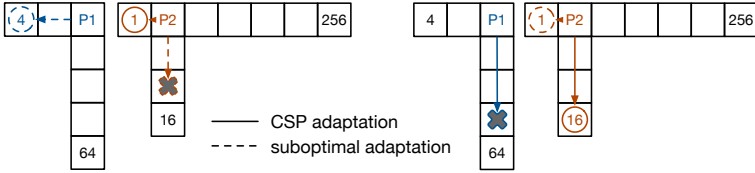

Figure 2: The didactic maze example. The solid line indicates the testing trajectory of adaptation policy produced by CSP, while the dashed line indicates sub-optimal adaptation trajectories: **(left)** BC-First in maze case 1 and **(right)** Self-Play in maze case 2.

**Case 1: Inadequacy of Non-Conservative Methods**   We show in this case that non-conservative algorithms easily overfit to an erroneous proxy model, lured to exploit "fake weaknesses" of the

---

[2]Rewards are set exponentially for efficient exploration of practical deep reinforcement learning algorithms.

Table 1: The training and test performances in maze environment **(left)** case 1 and **(right)** case 2.

| CASE 1 | Train | Test (against $\pi_B$) | CASE 2 | Train | Test (against $\pi_B$) |
|---|---|---|---|---|---|
| BC-First | 16 | -4 | BC-First | 16 | 16 |
| Self-Play | 1 | 1 | Self-Play | 1 | 1 |
| Dataset-Only | -64 | -64 | Dataset-Only | 16 | 16 |
| **CSP (Ours)** | 1 | 1 | **CSP (Ours)** | 16 | 16 |
| Oracle | 1 | 1 | Oracle | 16 | 16 |

opponent. Suppose that P1's real policy $\pi_B$ is as follows: (1) if P1 observes P2 $\rightarrow$ 16, P1 $\rightarrow$ 4; (2) if P1 observes P2 $\rightarrow$ 256, P1 $\rightarrow$ 64; (3) otherwise P1 does not move. This is a reasonably strong policy for P1 which tries to exploit P2. Suppose the dataset is collected with $\pi_B$ and P2's rollout policy is P2 $\rightarrow$ 256. Therefore, in the dataset, P1 is only observed going downwards. The proxy model trained by BC thus erroneously assumes that P1 $\rightarrow$ 64 under any conditions. The results are shown on the left side of Table 1. The BC-First method makes a disastrous generalization and responds with a P2 $\rightarrow$ 16 policy. Trying to exploit the "fake weaknesses" of the opponent results in failure in testing against the real opponent. However, CSP discovers that P2 $\rightarrow$ 16 is unsafe and learns to safely win the game by playing P2 $\rightarrow$ 1. Both CSP and max-min policy achieve the optimal solution. The Dataset-Only method is overly conservative and only gets -64. The trajectories of CSP and BC-First when exploiting the real target are shown on the left side of figure 2.

**Case 2: Performing Risk-Free Adaptation**    In this case, we show that CSP outperforms pure Self-Play (which does not exploit the opponent) by discovering risk-free exploitation opportunities. Suppose that P1 always plays an aggressive policy $\pi_B$: P1 $\rightarrow$ 64. The dataset $D$ is collected by $\pi_B$ with P2 playing P2 $\rightarrow$ 16 and P2 $\rightarrow$ 256 randomly. The results are shown on the right side of Table 1, and the trajectories of CSP and Self-Play are shown on the right side of Figure 2. The max-min policy produced by Self-Play cannot exploit P1's reckless policy. The generalization of the BC model is indeed correct in this specific situation, so BC-First is also optimal. Our algorithm recognizes that P2 $\rightarrow$ 16 is a risk-free exploitation because the dataset has witnessed that P1 still plays P1 $\rightarrow$ 64 even if P2 $\rightarrow$ 16 and successfully exploits the opponent.

It is noteworthy that we use the same hyperparameters for CSP in both cases and our algorithm learns different adaptation policies based solely on the different datasets used. It further validates that our algorithm is able to extract useful information from the dataset, and to perform risk-free adaptations.

## 6    Experiment

We conduct extensive experiments on the predator-prey in MPE [39], a competitive two-player MuJoCo environment [2, 10] that requires continuous control, and Google Football [21]. In all experiments, we compare CSP with (1) *BC-First*, and (2) *Self-Play*, which produces an approximate max-min conservative policy of the game. Self-Play can also be viewed as an ablation of CSP which does not utilize an offline dataset to adapt to targets. We also compare with multi-agent imitation learning (MAIL), and previous opponent exploitation method RNR [17], which does not perform conservative offline exploitation. More experiment details are available in Appendix G. We focus on performing opponent exploitation in competitive games. However, we also discuss the use of CSP for cooperative tasks in Appendix E.

### 6.1    Experiments in Predator-Prey and MuJoCo

In the predator-prey environment, there are three adversarial agents, one good agent, and two obstacles. The good agent is penalized for being hit by adversaries, whereas the adversaries are rewarded. In experiments presented in Table 2, our agent controls either the good agent or the adversaries, respectively. We use a pre-trained model as target $\pi_B$. We report the episode rewards in evaluation with $\pi_B$. We use three types of datasets (10R, 100R, 100S) to demonstrate the impact of dataset quality in offline adaptation. The prefix number indicates the number of trajectories in the dataset. Letter "R" indicates that the dataset is collected by $\pi_B$ playing with a random policy, while "S" indicates that the dataset is collected by $\pi_B$ playing with another well-trained opponent.

Table 2: Testing episode rewards of adaptation policy in predator-prey of MPE.

| | Controls Good Agent | | | Controls Adversaries | | |
|---|---|---|---|---|---|---|
| Dataset | 10R | 100R | 100S | 10R | 100R | 100S |
| **CSP(Ours)** | **-16.7±0.4** | **-17.5±0.4** | **-17.1±0.1** | **34.6±1.6** | **33.6±3.2** | **34.7±4.1** |
| BC-First | -29.8±4.0 | -21.8±1.6 | -18.5±0.4 | 21.3±0.8 | 25.2±0.9 | 27.8±1.9 |
| Self-Play | -17.6±1.5 | -17.6±1.5 | -17.6±1.5 | 29.7±2.8 | 29.7±2.8 | 29.7±2.8 |
| MAIL | -59.0±0.6 | -54.2±1.0 | -37.9±1.4 | 4.3±0.3 | 4.1±0.2 | 25.8±0.7 |
| Oracle | -16.0±0.3 | -16.0±0.3 | -16.0±0.3 | 34.3±6.2 | 34.3±6.2 | 34.3±6.2 |

Table 3: Experiments in MuJoCo YouShallNotPassHumans environment. We show the winning rates and episode rewards against 4 independent target opponents..

| | Method | 1 | 2 | 3 | 4 |
|---|---|---|---|---|---|
| Winning Rate | **CSP(Ours)** | **0.72 ± 0.06** | **0.65 ± 0.04** | **0.85 ± 0.08** | **0.88 ± 0.02** |
| | BC-First | 0.65 ± 0.13 | 0.46 ± 0.14 | 0.71 ± 0.17 | 0.72 ± 0.07 |
| | Self-Play | 0.54 ± 0.20 | 0.59 ± 0.14 | 0.52 ± 0.02 | 0.60 ± 0.19 |
| Episode Reward | **CSP(Ours)** | **878 ± 166** | **676 ± 100** | **1203 ± 209** | **1274 ± 73** |
| | BC-First | 693 ± 347 | 179 ± 456 | 848 ± 373 | 858 ± 185 |
| | Self-Play | 419 ± 527 | 500 ± 395 | 318 ± 106 | 601 ± 514 |

Intuitively, 10R has the lowest dataset quality, while 100S has the highest. We train the Oracle directly against $\pi_B$ as an upper bound. Self-Play and Oracle do not use datasets. MAIL imitates the dataset collection policy. It fails to learn meaningful behavior on datasets 10R and 100R where the dataset collection policy is a random policy. Table 2 shows that CSP outperforms baselines and achieves comparable performances with Oracle, regardless of the dataset quality. CSP effectively learns to exploit the opponent safely, while the non-conservative method BC-First fails to do so. Moreover, BC-First is significantly more sensitive to dataset quality. In the controlling adversaries scenario, even with the 100S dataset, BC-First still performs worse than Self-Play, which does not exploit at all. It shows that the non-conservative method overfits the proxy model severely.

The YouShallNotPassHumanoids environment [2, 10] is a MuJoCo-based two-player competitive game where the runner's objective is to pass the blocker and reach the red line on the opposite side. In our experiments, our agent acts as the runner, and the blocker is the target opponent for exploitation. In Table 3, we use four independently pre-trained blocker models as opponent targets. Since the exploitability of different opponent models can vary significantly, results against different opponents are not directly comparable. So we report results for all targets. CSP successfully exploits the opponent, and significantly outperforms the baselines for all targets. Additionally, we include a visualization of CSP's learned behavior in Appendix G.

## 6.2 Experiments in Google Football

Google Football [21] is a popular and challenging benchmark for MARL. In our experiments, the adaptation policy controls all players in one team, while the opponent target policy controls the other team. We conduct experiments in 4 scenarios: *academy_3_vs_1_with_keeper* (3vs1, where our agent acts as either the defender or the attacker), *academy_run_pass_and_shoot_with_keeper* (RPS, defender), and *academy_counterattack_easy* (defender). We report the winning rates of adaptation policies for five independently pre-trained opponent targets.

The experiment results in Table 4 show that our algorithm CSP outperforms baselines for almost all targets in all scenarios. Notably, the BC-First algorithm even

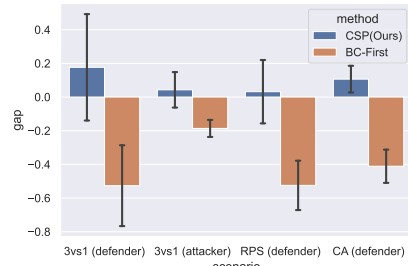

Figure 3: The average test-train performance gap in four scenarios of Google Football. A negative gap indicates the occurrence of unsafe exploitation.

Table 4: Winning rates of offline adaptation policy in 4 scenarios of Google Football: 3vs1 (defender), 3vs1 (attacker), RPS (defender), and Counterattack Easy (defender). For each scenario, we experiment with 5 independent target opponents.

| Scenario | Method | 1 | 2 | 3 | 4 | 5 |
|---|---|---|---|---|---|---|
| 3vs1 defender | **CSP(Ours)** | **0.9 ± 0.06** | **0.6 ± 0.12** | **0.45 ± 0.04** | **0.32 ± 0.11** | **0.76 ± 0.16** |
| | BC-First | 0.64 ± 0.04 | 0.46 ± 0.09 | 0.2 ± 0.01 | 0.16 ± 0.04 | 0.56 ± 0.09 |
| | Self-Play | 0.29 ± 0.09 | 0.34 ± 0.1 | 0.26 ± 0.16 | 0.29 ± 0.18 | 0.3 ± 0.13 |
| 3vs1 attacker | **CSP(Ours)** | 0.81 ± 0.14 | **0.88 ± 0.02** | **0.83 ± 0.07** | **0.84 ± 0.05** | **0.78 ± 0.08** |
| | BC-First | **0.83 ± 0.03** | 0.74 ± 0.21 | 0.68 ± 0.07 | 0.79 ± 0.11 | 0.71 ± 0.15 |
| | Self-Play | 0.75 ± 0.15 | 0.76 ± 0.08 | 0.73 ± 0.13 | 0.7 ± 0.21 | 0.74 ± 0.08 |
| RPS defender | **CSP(Ours)** | 0.51 ± 0.33 | **0.71 ± 0.07** | **0.56 ± 0.13** | **0.38 ± 0.07** | **0.79 ± 0.09** |
| | BC-First | 0.51 ± 0.24 | 0.41 ± 0.07 | 0.34 ± 0.18 | **0.38 ± 0.04** | 0.71 ± 0.02 |
| | Self-Play | **0.57 ± 0.14** | 0.36 ± 0.04 | 0.25 ± 0.04 | **0.37 ± 0.1** | 0.76 ± 0.03 |
| Counter-attack defender | **CSP(Ours)** | **0.93 ± 0.02** | **0.88 ± 0.04** | **0.81 ± 0.25** | **0.81 ± 0.02** | **0.75 ± 0.06** |
| | BC-First | 0.7 ± 0.17 | 0.52 ± 0.07 | 0.53 ± 0.09 | 0.69 ± 0.14 | 0.5 ± 0.09 |
| | Self-Play | 0.55 ± 0.13 | 0.41 ± 0.1 | 0.46 ± 0.09 | 0.36 ± 0.08 | 0.36 ± 0.07 |

performs worse than Self-Play, which does not utilize the opponent's information to exploit. It indicates that previous non-conservative algorithms are not guaranteed to use the opponent's information safely in offline adaptation.

In Figure 3, we show the gap of performances between testing and training: $J(\pi, \pi_B) - J(\pi, \mu)$. A negative gap indicates that although $\pi$ learns to exploit $\mu$ during training, such exploitation is risky and can not be guaranteed in testing. The results demonstrate the significant challenges that offline policy adaptation poses for baseline exploitation algorithms. BC-First overfits the proxy model in training and experiences a severe performance drop in testing. Conversely, CSP has a positive gap on average, indicating its ability to find conservative exploitation opportunities. Because CSP optimizes for the worst-case performance, it tends to produce a proxy model that is possibly stronger than the real target. Thus, its evaluation performances are even higher than those in training.

## 6.3 Comparison with Non-Conservative Opponent Exploitation

We compare CSP with the previous safe exploitation method RNR [17], which minimizes the distance to NE when exploiting. RNR assumes that the opponent plays an NE policy with probability $1 - p$ and follows an estimated opponent model with $p$. It then learns an exploitation policy. However, RNR does not optimize the conservative offline adaptation objective, and it still fails to handle offline policy adaptation due to the neglect of possible errors in the estimated opponent model. In Figure 4, the results show that CSP outperforms RNR for every dataset and every value of $p$. RNR(0) is equivalent to Self-Play, and RNR(1) is equivalent to BC-First. Results for these two have already been reported in Table 2. The results further support our claim that previous methods struggle with offline policy adaptation and highlight the efficiency and significance of our algorithm.

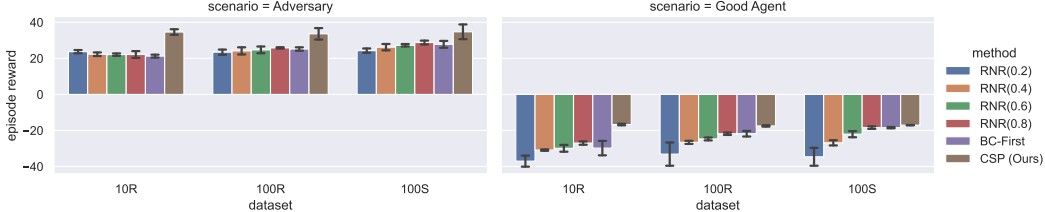

Figure 4: Comparison with RNR [17] in predator-prey, where our agent controls **(left)** adversaries and **(right)** good agent respectively. The y-axis represents the episode reward in testing.

# 7 Conclusion

We provide a thorough analysis of the unique challenges in offline policy adaptation, which are often neglected by previous literature. We propose the novel learning objective of conservative offline adaptation that optimizes the worst-case performance against any dataset-consistent opponent proxy. We propose an efficient algorithm that learns near-optimal conservative adaptation policies. Extensive empirical results in Maze, predator-prey, MuJoCo, and Google Football demonstrate that our algorithm significantly outperforms baselines.

One limitation of our work is that optimizing worst-case performance in real-world applications may not always be necessary since participants are not fully rational in practice. One future direction to settle this problem is to extend offline adaptation to non-stationary opponents, which is orthogonal to our current study. Those methods typically focus on adapting policy according to inferred opponent information in evaluation, while our algorithm prevents performance degradation in testing against a consistent opponent. One way is to use our algorithm to train a population of risk-free exploitation policies against opponents of different types and adapt exploitation policy in testing according to the opponent's posterior distribution. We discuss potential negative social impacts in Appendix F.

## Acknowledgements and Disclosure of Funding

The authors would like to thank the anonymous reviewers for their insightful discussions and helpful suggestions. This work is supported in part by Science and Technology Innovation 2030 – "New Generation Artificial Intelligence" Major Project (No. 2018AAA0100904) and National Natural Science Foundation of China (62176135).

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

# A Proof

## Proof of Proposition 4.2

**Proposition 4.2** The performance gap of evaluating policy profile $(\pi, \mu)$ and $(\pi, \pi_B)$ at state $s$ is $\varepsilon(s) = V^{\pi,\pi_B}(s) - V^{\pi,\mu}(s)$, which can be decomposed in a Bellman-like recursion:

$$
\begin{aligned}
\varepsilon(s) &= V^{\pi,\pi_B}(s) - V^{\pi,\mu}(s) \\
&= \sum_{a_1, a_2, s'} \pi(a_1|s) P_M(s'|s, \boldsymbol{a}) (\pi_B(a_2|s) - \mu(a_2|s)) (r(s, \boldsymbol{a}, s') + \gamma V^{\pi,\mu}(s')) \\
&\quad + \sum_{a_1, a_2, s'} \pi(a_1|s) \pi_B(a_2|s) P_M(s'|s, \boldsymbol{a}) \gamma \varepsilon(s')
\end{aligned}
\tag{7}
$$

*Proof.*

$$
\begin{aligned}
\varepsilon(s) &= V^{\pi,\pi_B}(s) - V^{\pi,\mu}(s) \\
&= \sum_{a_1} \pi(a_1|s) \sum_{a_2} \pi_B(a_2|s) \sum_{s'} P_M(s'|s, \boldsymbol{a}) \left[ r(s, \boldsymbol{a}, s') + \gamma V^{\pi,\mu}(s') + \gamma \varepsilon(s') \right] \\
&\quad - \sum_{a_1} \pi(a_1|s) \sum_{a_2} \mu(a_2|s) \sum_{s'} P_M(s'|s, \boldsymbol{a}) \left[ r(s, \boldsymbol{a}, s') + \gamma V^{\pi,\mu}(s') \right] \\
&= \sum_{a_1} \pi(a_1|s) \sum_{a_2} \left[ (\pi_B(a_2|s) - \mu(a_2|s)) A + B \right],
\end{aligned}
\tag{8}
$$

where

$$
A = \sum_{s'} P_M(s'|s, \boldsymbol{a}) r(s, \boldsymbol{a}, s'),
\tag{9}
$$

and

$$
\begin{aligned}
B &= \pi_B(a_2|s) \sum_{s'} P_M(s'|s, \boldsymbol{a}) \gamma \left( V^{\pi,\mu}(s') + \varepsilon(s') \right) - \mu(a_2|s) \sum_{s'} P_M(s'|s, \boldsymbol{a}) \gamma V^{\pi,\mu}(s') \\
&= \sum_{s'} P_M(s'|s, \boldsymbol{a}) \left[ \pi_B(a_2|s) \gamma \left( V^{\pi,\mu}(s') + \varepsilon(s') \right) - \mu(a_2|s) \gamma V^{\pi,\mu}(s') \right] \\
&= \sum_{s'} P_M(s'|s, \boldsymbol{a}) \left[ \gamma V^{\pi,\mu}(s') (\pi_B(a_2|s) - \mu(a_2|s)) + \gamma \pi_B(a_2|s) \varepsilon(s') \right].
\end{aligned}
\tag{10}
$$

Putting equation 9 and 10 back, we have

$$
\begin{aligned}
\varepsilon(s) &= V^{\pi,\pi_B}(s) - V^{\pi,\mu}(s) \\
&= \sum_{a_1} \pi(a_1|s) \sum_{a_2} \left[ (\pi_B(a_2|s) - \mu(a_2|s)) \sum_{s'} P_M(s'|s, \boldsymbol{a}) r(s, \boldsymbol{a}, s') + \right. \\
&\quad \left. \sum_{s'} P_M(s'|s, \boldsymbol{a}) \left[ \gamma V^{\pi,\mu}(s') (\pi_B(a_2|s) - \mu(a_2|s)) + \gamma \pi_B(a_2|s) \varepsilon(s') \right] \right] \\
&= \sum_{a_1, a_2, s'} \pi(a_1|s) P_M(s'|s, \boldsymbol{a}) (\pi_B(a_2|s) - \mu(a_2|s)) (r(s, \boldsymbol{a}, s') + \gamma V^{\pi,\mu}(s')) \\
&\quad + \sum_{a_1, a_2, s'} \pi(a_1|s) \pi_B(a_2|s) P_M(s'|s, \boldsymbol{a}) \gamma \varepsilon(s').
\end{aligned}
\tag{11}
$$

$\square$

## Proof of Theorem 4.3

**Theorem 4.3** We use $\varepsilon = \sum_{s_0} p_0(s_0) \varepsilon(s_0)$ to denote the overall performance gap between policy profile $(\pi, \mu)$ and $(\pi, \pi_B)$. In any 2-player fully observable game, for all reward functions, $\varepsilon = 0$ if and only if $\pi_B(a|s) = \mu(a|s), \forall s$, s.t. $d_{\pi,\pi_B}(s) > 0$.

*Proof. Sufficiency.* Whenever $\pi_B(a|s) = \mu(a|s)$ holds, the first term in equation 1 is 0. If $\forall s$ s.t. $d_{\pi,\pi_B}(s) > 0$, $\pi_B(a|s) = \mu(a|s)$, according to Proposition 4.2, by expanding the expression for $\varepsilon(s)$ recursively, we have for $\forall s$ s.t. $d_{\pi,\pi_B}(s) > 0$, $\varepsilon(s) = 0$. Therefore, $\varepsilon = \sum_{s_0} p(s_0)|\varepsilon(s_0)| = 0$.

*Necessity.* We first show by contradiction that $\varepsilon = \sum_{s_0} p(s_0)\varepsilon(s_0)$ requires $\forall s$ s.t. $p_0(s) > 0$, $\pi_B(a|s) = \mu(a|s)$. If $\pi_B(a|s) \neq \mu(a|s)$, according to Proposition 4.2, we can change the reward function $r(s, \boldsymbol{a}, s')$ to change the value of $\varepsilon(s)$, because the reward function is arbitrary. So $\varepsilon = \sum_{s_0} p(s_0)\varepsilon(s_0)$ does not hold anymore, and this is a contradiction. Therefore, $\varepsilon(s) = \sum_{s_0,s_1} p(s_0)\pi(a_1|s_0)\pi_B(a_2|s_0)P_M(s_1|s,\boldsymbol{a})\varepsilon(s_1) = \mathbb{E}_{s_1 \sim (\pi,\pi_B)}[\varepsilon(s_1)]$. Using the same arguments, it can be shown that $\forall s$, s.t. $Pr(s_1 = s) > 0$, $\pi_B(a|s) = \mu(a|s)$. By expanding recursively, the statement is proved. $\square$

**Proof of Theorem 4.7**

We first prove a Lemma.

**Lemma A.1.** *In a two-player game, suppose that $\pi$ is player 1's policy, $\alpha$ and $\mu$ are player 2's policies. We use $\pi_{(\pi,\alpha)}^{joint}$ and $\pi_{(\pi,\mu)}^{joint}$ to denote the joint policy profiles $(\pi, \alpha)$ and $(\pi, \mu)$. Then we have $D_{\mathrm{KL}}\left(\pi_{(\pi,\alpha)}^{joint}(\cdot|s)\|\pi_{(\pi,\mu)}^{joint}(\cdot|s)\right) = D_{\mathrm{KL}}\left(\alpha(\cdot|s)\|\mu(\cdot|s)\right)$, where $D_{\mathrm{KL}}$ denotes KL divergence.*

*Proof.* According to definition,

$$
\begin{aligned}
D_{\mathrm{KL}}\left(\pi_{(\pi,\alpha)}^{\text{joint}}(\cdot|s)\|\pi_{(\pi,\mu)}^{\text{joint}}(\cdot|s)\right) &= \sum_{a_1,a_2} \pi(a_1|s)\alpha(a_2|s) \log \frac{\pi(a_1|s)\alpha(a_2|s)}{\pi(a_1|s)\mu(a_2|s)} \\
&= \sum_{a_1} \pi(a_1|s) \sum_{a_2} \alpha(a_2|s) \log \frac{\alpha(a_2|s)}{\mu(a_2|s)} \\
&= \sum_{a_1} \pi(a_1|s) D_{\mathrm{KL}}\left(\alpha(\cdot|s)\|\mu(\cdot|s)\right) \\
&= D_{\mathrm{KL}}\left(\alpha(\cdot|s)\|\mu(\cdot|s)\right).
\end{aligned}
\tag{12}
$$

$\square$

Additionally, we use the Theorem 1 in [36] and we provide a restatement. The original paper [36] minimizes accumulated discounted cost. In contrast, an agent maximizes its return in our paper.

**Theorem A.2.** *(Theorem 1 in [36]) Let $\epsilon = \max_s |\mathbb{E}_{a \sim \pi_2(a|s)}[A^{\pi_1}(s,a)]|$, then*

$$
J(\pi_2) - J(\pi_1) \geq \mathbb{E}_{s \sim d_{\pi_1}(s)} \mathbb{E}_{a \sim \pi_2(a|s)}\left[A^{\pi_1}(s,a)\right] - \frac{2\epsilon\gamma}{(1-\gamma)^2} \max_s D_{\mathrm{KL}}\left(\pi_1(\cdot|s)\|\pi_2(\cdot|s)\right). \tag{13}
$$

Finally, we prove Theorem 4.7.

**Theorem 4.7** In a two-player game, suppose that $\pi$ is player 1's policy, $\alpha$ and $\mu$ are player 2's policies. Assume that $\max_s D_{\mathrm{KL}}\left(\alpha(\cdot|s)\|\mu(\cdot|s)\right) \leq \delta$. Let $R_M$ be the maximum magnitude of return obtainable for player 1 in this game, and let $\gamma$ be the discount factor. We use $J(\pi, \mu)$ and $J(\pi, \alpha)$ to denote the return for player 1 when playing policy profiles $(\pi, \mu)$ and $(\pi, \mu)$ respectively. Then,

$$
J(\pi, \mu) - J(\pi, \alpha) \geq -R_M\sqrt{2\delta}\left(1 + \frac{2\gamma\delta}{(1-\gamma)^2}\right). \tag{14}
$$

*Proof.* According to Theorem A.2, we have

$$
\begin{aligned}
J(\pi, \mu) - J(\pi, \alpha) &\geq \mathbb{E}_{s \sim d_{(\pi,\alpha)}} \mathbb{E}_{a_1 \sim \pi, a_2 \sim \mu}[A^{(\pi,\alpha)}(s,\boldsymbol{a})] \\
&\quad - \frac{2\epsilon\gamma}{(1-\gamma)^2} \max_s D_{\mathrm{KL}}\left(\pi_{(\pi,\alpha)}^{\text{joint}}(\cdot|s)\|\pi_{(\pi,\mu)}^{\text{joint}}(\cdot|s)\right),
\end{aligned}
\tag{15}
$$

where $\epsilon = \max_s |\mathbb{E}_{a_1 \sim \pi, a_2 \sim \mu}[A^{(\pi,\alpha)}(s, \boldsymbol{a})]| \geq 0$. Use Lemma A.1, we have

$$
\begin{aligned}
J(\pi, \mu) - J(\pi, \alpha) &\geq \mathbb{E}_{s \sim d_{(\pi,\alpha)}} \mathbb{E}_{a_1 \sim \pi, a_2 \sim \mu}[A^{(\pi,\alpha)}(s, \boldsymbol{a})] - \frac{2\epsilon\gamma}{(1-\gamma)^2} \max_s D_{\mathrm{KL}}\left(\alpha(\cdot|s)\|\mu(\cdot|s)\right) \\
&\geq -\epsilon - \frac{2\epsilon\gamma}{(1-\gamma)^2} \max_s D_{\mathrm{KL}}\left(\alpha(\cdot|s)\|\mu(\cdot|s)\right) \\
&= -\epsilon \left(1 + \frac{2\gamma}{(1-\gamma)^2} \max_s D_{\mathrm{KL}}\left(\alpha(\cdot|s)\|\mu(\cdot|s)\right)\right) \\
&\geq -\epsilon \left(1 + \frac{2\gamma\delta}{(1-\gamma)^2}\right).
\end{aligned}
$$

(16)

Next, we use KL divergence to bound $\epsilon$. According to definition, $\left|Q^{(\pi,\alpha)}(s, \boldsymbol{a})\right| \leq R_M$. So we have,

$$
\begin{aligned}
\epsilon &= \max_s |\mathbb{E}_{a_1 \sim \pi, a_2 \sim \mu}[A^{(\pi,\alpha)}(s, \boldsymbol{a})]| \\
&= \max_s \left| \sum_{a_1} \pi(a_1|s) \sum_{a_2} \mu(a_2|s) \left(Q^{(\pi,\alpha)}(s, \boldsymbol{a}) - V^{(\pi,\alpha)}(s)\right) \right| \\
&= \max_s \left| \sum_{a_1} \pi(a_1|s) \sum_{a_2} (\mu(a_2|s) - \alpha(a_2|s)) Q^{(\pi,\alpha)}(s, \boldsymbol{a}) \right| \\
&\leq \max_s \sum_{a_1} \pi(a_1|s) \sum_{a_2} |\mu(a_2|s) - \alpha(a_2|s)| |Q^{(\pi,\alpha)}(s, \boldsymbol{a})| \\
&\leq \max_s \sum_{a_1} \pi(a_1|s) \sum_{a_2} |\mu(a_2|s) - \alpha(a_2|s)| R_M
\end{aligned}
$$

(17)

With Pinsker's inequality [4], the total variation distance is bounded by KL divergence:

$$
\delta(\alpha(\cdot|s), \mu(\cdot|s)) = \frac{1}{2}\|\alpha(\cdot|s) - \mu(\cdot|s)\|_1 \leq \sqrt{\frac{1}{2} D_{\mathrm{KL}}\left(\alpha(\cdot|s)\|\mu(\cdot|s)\right)}.
$$

(18)

Therefore,

$$
\begin{aligned}
\epsilon &\leq \max_s \sum_{a_1} \pi(a_1|s) \cdot 2R_M \delta(\alpha(\cdot|s), \mu(\cdot|s)) \\
&\leq \sqrt{2} R_M \cdot \max_s \sqrt{D_{\mathrm{KL}}\left(\alpha(\cdot|s)\|\mu(\cdot|s)\right)} \\
&\leq R_M \sqrt{2\delta}.
\end{aligned}
$$

(19)

Put inequality 19 back into 16, and we get

$$
J(\pi, \mu) - J(\pi, \alpha) \geq -R_M \sqrt{2\delta} \left(1 + \frac{2\gamma\delta}{(1-\gamma)^2}\right).
$$

(20)

$\square$

### Proof of Theorem 4.8

**Theorem 4.8** Let $\pi^*$ be the optimal risk-free offline adaptation policy at the convergence of the optimization of objective 2, and let $\tilde{\pi}$ be the policy at the convergence of objective 5. Then the worst-case adaptation performance of $\tilde{\pi}$ is near-optimal:

$$
\min_{\mu \in \mathcal{C}_D} J(\pi^*, \mu) \geq \min_{\mu \in \mathcal{C}_D} J(\tilde{\pi}, \mu) \geq \min_{\mu \in \mathcal{C}_D} J(\pi^*, \mu) - R_M \sqrt{2\delta} \left(1 + \frac{2\gamma\delta}{(1-\gamma)^2}\right).
$$

(21)

*Proof.* According to definition, $\pi^*$ is the solution to the optimization objective 2, so $\min_{\mu \in \mathcal{C}_D} J(\pi^*, \mu) \geq \min_{\mu \in \mathcal{C}_D} J(\tilde{\pi}, \mu)$.

Suppose that $(\pi^*, \mu^*)$ and $(\tilde{\pi}, \tilde{\alpha})$ are the solutions to objectives 2 and 5 respectively. For $\forall \mu \in \mathcal{C}_D$, let $\mathcal{F}(\mu) = \{\alpha | \forall s \notin D, \alpha(\cdot|s) = \mu(\cdot|s); \max_{s \in D} D_{\mathrm{KL}}(\pi_B(\cdot|s)\|\alpha(\cdot|s)) \leq \delta\}$ be the set of corresponding $\alpha$ policies which are identical to $\mu$ on OOD states. Observe that $\max_s D_{\mathrm{KL}}(\pi_B(\cdot|s)\|\alpha(\cdot|s)) \leq \delta$ also holds. Therefore, according to Theorem 4.7, $\forall \pi, \forall \mu \in$

$\mathcal{C}_D, \forall \alpha \in \mathcal{F}(\mu), J(\pi, \alpha) \geq J(\pi, \mu) - R_M\sqrt{2\delta}\left(1 + \frac{2\gamma\delta}{(1-\gamma)^2}\right)$. Taking the minimization over $\mu$ and $\alpha$, we get $\forall \pi, \min_\mu \min_{\alpha \in \mathcal{F}(\mu)} J(\pi, \alpha) \geq \min_\mu J(\pi, \mu) - R_M\sqrt{2\delta}\left(1 + \frac{2\gamma\delta}{(1-\gamma)^2}\right)$. Observe that the left part is equivalent to $\min_{\alpha:\max_{s\in D} D_{\mathrm{KL}}(\alpha(\cdot|s)\|\pi_B(\cdot|s))\leq\delta} J(\pi, \alpha)$. Taking the maximization over $\pi$ for both sides, we get $J(\tilde{\pi}, \tilde{\alpha}) \geq J(\pi^*, \mu^*) - R_M\sqrt{2\delta}\left(1 + \frac{2\gamma\delta}{(1-\gamma)^2}\right)$.

Let $J(\tilde{\pi}, \mu') = \min_{\mu \in \mathcal{C}_D} J(\tilde{\pi}, \mu)$. Observe that $\mu'$ also satisfies the KL divergence constraint: $\max_{s\in D} D_{\mathrm{KL}}(\pi_B(\cdot|s)\|\mu'(\cdot|s)) \leq \delta$, and $(\tilde{\pi}, \tilde{\alpha})$ is the optimal solution to objective 5, so $J(\tilde{\pi}, \mu') \geq J(\tilde{\pi}, \tilde{\alpha})$. So we get $\min_{\mu \in \mathcal{C}_D} J(\tilde{\pi}, \mu) \geq J(\tilde{\pi}, \tilde{\alpha}) \geq J(\pi^*, \mu^*) - R_M\sqrt{2\delta}\left(1 + \frac{2\gamma\delta}{(1-\gamma)^2}\right)$.  □

## B   Related Work

**RL in multi-agent games.** Multi-agent reinforcement learning (MARL) under the centralized training decentralized execution (CTDE) paradigm have shown promising performances in learning coordination behavior for cooperative multi-agent tasks such as SMAC [35] and Google Football [21]. Representative works consist of policy gradient algorithms [6, 48], and value decomposition methods [34, 40]. There also have been extensive research in competitive environments such as Texas hold'em and MOBA games. Some works design theoretically sounded RL algorithms which are guaranteed to converge to approximate NE [3]. Some works propose RL methods based on self-play [45, 46, 50] to train max-min policies which show strong empirical performances.

**Offline RL.** Offline reinforcement learning [22] learns purely from batched data. Representative works take a conservative view of out-of-distribution state-action pairs to mitigate the distributional shift problem. BCQ [8] uses VAE to propose candidate actions that are within the support of dataset. IQL [19] uses expectile regression to learn Q function to avoid querying out-of-distribution actions. CQL [20] puts regularization on the learning of Q function to penalize out-of-distribution actions.

**Opponent exploitation.** There have been extensive study on various aspects of opponent exploitation [1, 28]. Some employ deep reinforcement learning to exploit the opponent in continuous control tasks using policy gradients estimated from direct interaction [2, 10, 12]. Some works, including RNR [17] and SES [24] study safe exploitation given an estimated opponent model. They keep the exploitation policy close to NE in order to minimize the loss if the opponent changes its policy adversarially. Some work, for example, GSCU [7], studies zero-shot generalization ability to exploit totally unknown opponents. To our best knowledge, there lacks a thorough investigation into the offline adaptation problem.

**Robust RL.** Formulated as robust-MDP (RMDP), robust RL [32, 30] is dedicated to learn a robust policy against perturbation in environment dynamics, e.g., observation noise, action delay. Formally, the unknown transition model in testing environment lies in an uncertainty set $\mathcal{P}$ which contains all models within a certain distance from the training model. Robust RL solves a max-min problem, optimizing the worst-case performance against any transition model in $\mathcal{P}$. Some works uses adversarial training [32, 16]. RORL [42] uses smoothing on the Q function to learn robust policy. The similarity to our work is that, our offline adaptation problem also optimizes the worst-case performance against the target's behavior on out-of-distribution states. However, we do not have any distance constraints on the target's policy on out-of-distribution states. To our best knowledge, our paper is the first to investigate into the offline adaptation problem.

## C   Algorigthm

The CSP algorithm is illustrated in Algorithm 1. The proxy model is trained adversarially against our agent, therefore, we set the proxy's reward function to be the negative of our agent's reward. In our experiments on an $n$-player environment, we assume that we control $n_1$ players, while the target controls $n_2$ players, and $n_1 + n_2 = n$. Therefore, we use MAPPO [48] as the basic learning algorithm for players of both sides, since players from the same side are fully cooperative. MAPPO deals with the cooperative multi-agent reinforcement learning problem through learning a centralized value function conditioned on global state, and a decentralized control policy conditioned on local observations. We use self-play with alternative update to learn both adaptation policy $\pi$, and target's

proxy model $\mu$ simultaneously. We use a soft behavior cloning regularization term to minimize the KL-divergence between proxy model $\mu$ and target policy $\pi_B$.

---

**Algorithm 1** Constrained Self-Play (CSP)

---

**Input:** dataset $D$, environment $env$, learning rate $\alpha$, regularization coefficient $C_{\mathrm{BC}}$
**Output:** adaptation policy $\pi$, target's policy proxy $\mu$
1: Initialize adaptation policy $\pi$, critic $v^{\pi}$; and target's policy proxy $\mu$, critic $v^{\mu}$
2: **while** not converged **do**
3:     Collect trajectories $\{\mathcal{T}_i\} \leftarrow \mathrm{rollout}(\pi, \mu; env)$
4:     **if** our turn **then**
5:         For each trajectory $\mathcal{T}_i$, calculate return target $R_t^{\gamma}$ and advantage $\hat{A}_t$ of ***our side*** using GAE with value function $v^{\pi}$ for each step $t$
6:         $v^{\pi} \leftarrow v^{\pi} - \alpha \nabla_{v^{\pi}} \frac{1}{|\cup \mathcal{T}_i|} \sum_{(s_t, R_t^{\gamma}) \sim \cup \mathcal{T}_i} (R_t^{\gamma} - v^{\pi}(s_t))$
7:         $\pi \leftarrow \pi + \alpha \nabla_{\pi} \frac{1}{|\cup \mathcal{T}_i|} \sum_{(o_t, a_t, \pi_{old}, \hat{A}_t) \sim \cup \mathcal{T}_i} \min \left( \frac{\pi(a_t|o_t)}{\pi_{old}(a_t|o_t)} \hat{A}_t, \right.$
8:             $\left. \mathrm{clip} \left( \frac{\pi(a_t|o_t)}{\pi_{old}(a_t|o_t)}, 1 - \epsilon, 1 + \epsilon \right) \hat{A}_t \right)$
9:     **else**
10:        For each trajectory $\mathcal{T}_i$, calculate return target $R_t^{\gamma}$ and advantage $\hat{A}_t$ of ***target agent side*** using GAE with value function $v^{\mu}$ for each step $t$
11:        $\mathcal{B} \leftarrow$ sample a random batch from $D$
12:        $v^{\mu} \leftarrow v^{\mu} - \alpha \nabla_{v^{\mu}} \frac{1}{|\cup \mathcal{T}_i|} \sum_{(s_t, R_t^{\gamma}) \sim \cup \mathcal{T}_i} (R_t^{\gamma} - v^{\mu}(s_t))$
13:        $\mu \leftarrow \mu + \alpha \nabla_{\mu} \left\{ \frac{1}{|\cup \mathcal{T}_i|} \sum_{(o_t, a_t, \pi_{old}, \hat{A}_t) \sim \cup \mathcal{T}_i} \min \left( \frac{\mu(a_t|o_t)}{\mu_{old}(a_t|o_t)} \hat{A}_t, \right. \right.$
14:             $\left. \mathrm{clip} \left( \frac{\mu(a_t|o_t)}{\mu_{old}(a_t|o_t)}, 1 - \epsilon, 1 + \epsilon \right) \hat{A}_t \right) - C_{\mathrm{BC}} \cdot \frac{1}{|\mathcal{B}|} \sum_{(o,a) \in \mathcal{B}} -\log \mu(a|o) \right\}$
15:     **end if**
16: **end while**

---

# D   Experiment on Didactic Maze

We show experiment details of the Maze example in this section. The learning curve in case 1 is shown in Figure 5. In the BC-First algorithm, since the dataset only contains trajectories that P1 $\rightarrow$ 64, the BC model makes a wrong and risky generalization that assumes P1 always goes downwards to 64 no matter how P2 acts. Therefore, the P2 policy learns to exploit this "weakness" by playing P2 $\rightarrow$ 16, and achieves 16 in training phase. However, during testing, P2 obtains -4 reward through trying to exploit an nonexistent weakness of P1.

Our algorithm avoids the trap of imaginary weakness, and learns to safely win the game, while the BC-First method makes risky exploitations which will not work in evaluation finally. Our algorithm keeps conservative for states outside dataset, and admits the possibility that P1 could go leftwards to win the game if P2 does not plays P2 $\rightarrow$ 1.

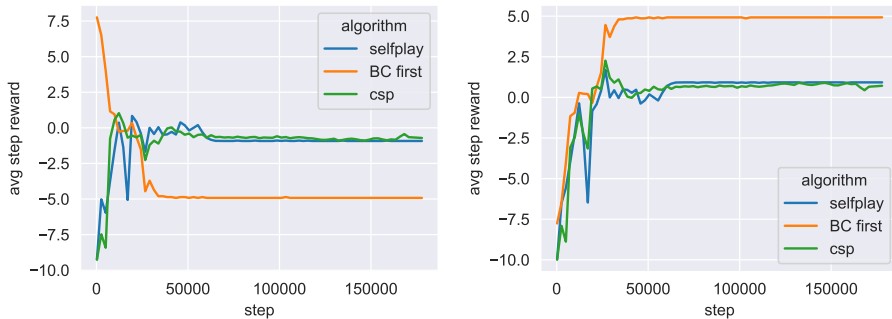

Figure 5: The training curve of amortized average per step reward in maze game case 1 in **training** (against the proxy). The x-axis represents training steps. **Left.** Average per step reward for P1. **Right.** Average per step reward for P2.

The learning curve in case 2 is shown in Figure 6. As can be seen in Figure 6, our algorithm quickly learns to exploit while the max-min strategy produced by Self-Play fails to.

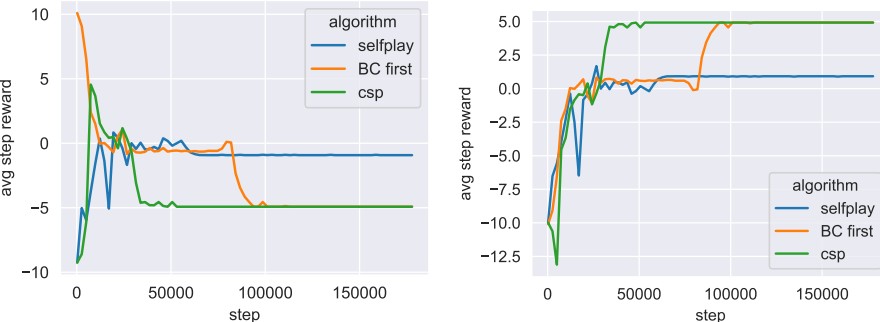

Figure 6: The training curve of amortized average per step reward in maze game case 2 in **training** (against the proxy). The x-axis represents training steps. **Left.** Average per step reward for P1. **Right.** Average per step reward for P2.

We use the same set of hyper-parameters (including the same coefficient $C_{BC}$ for the BC regularization term) for these two cases. The hyper-parameters are shown in Table 5. The dataset $D_1 = \{\text{trajectories}(P1 \rightarrow 64, P2 \rightarrow 256)\}$ in case 1, and our algorithm learns to play $P2 \rightarrow 1$; while in case 2, given dataset $D_2 = \{\text{trajectories}(P1 \rightarrow 64, P2 \rightarrow 256), \text{trajectories}(P1 \rightarrow 64, P2 \rightarrow 16)\}$, our algorithm learns to play $P2 \rightarrow 16$. We use the same set of hyper-parameters for both cases. Therefore, our algorithm produces different policies simply because of different datasets given. It further validates that our algorithm is able to extract useful information from dataset, and only perform risk-free adaptations.

Table 5: Hyper-parameters for maze.

| ppo_epoch | 1 | num_mini_batch | 1 | entropy_coef | 0.3 |
|---|---|---|---|---|---|
| use_gae | True | gamma | 0.99999 | gae_lambda | 0.95 |
| critic_lr | 7e-4 | lr | 7e-4 | weight_decay | 0 |
| adam_eps | 1e-5 | n_rollout_threads | 20 | ppo_episode_length | 12 |
| data_chunk_length | 12 | steps | 1.8K | max_grad_norm | 0.5 |
| bc_regularization_coef | 10 | bc_batch_size | 8 | network | MLP |

## E   Conservative Offline Adaptation for Cooperative Tasks

We evaluate our method in competitive games to perform opponent exploitation. Nevertheless, our method is also applicable to cooperative environments. The objective of CSP in cooperative games maintains the same min-max structure because it promotes adherence to the policy exposed by dataset for states within the dataset, while aiming to optimize for worst-case performance for states not in the dataset. Therefore, in cooperative games, the policy regularization term incentivizes the adaptation agent to collaborate with the target agent on states within the dataset, and the minimization over opponent proxy $\mu$ encourages the adaptation agent to be conservative and not to rely on the target agent's OOD policy. Note that in objective 5 and Algorithm 1, no matter whether the game is competitive or cooperative, the minimization over $\mu$ in training can be achieved by setting the target agent's reward function to be the negative of our adaptation agent's reward function. We then train $\mu$ to maximize this reward with policy regularization in self-play.

Although our method is applicable to cooperative environments as well, the challenges caused by offline policy adaptation are more significant in competitive games. Recall that conservative offline adaptation optimizes

$$\max_{\pi} \min_{\mu} J(\pi, \mu), \ s.t. \ \mu \in \mathcal{C}_D. \tag{22}$$

For cooperative tasks, it requires that the teammate $\mu$ will not cooperate on states outside dataset in cooperative games. Therefore, for the adaptation policy, staying within dataset could be a trivial and

near-optimal solution for most cooperative problems if we would like to maximize the worst-case performance against any dataset-consistent teammate.

## F   Potential Negative Social Impacts

Inappropriate use of exploitation algorithms may result in negative social impacts. An example of its negative impact is the exacerbation of inequality in society. Companies who have access to large amounts of data can utilize their customers more effectively. However, our algorithm is general-purpose and depends on pre-collected data. We advocate for strict laws and regulations that protect user privacy data to avoid negative impacts. It is recommended that products utilizing exploitation algorithms are made public and supervised.

## G   Experiment Details

### G.1   Environments

The experiment environments are illustrated in figure 7.

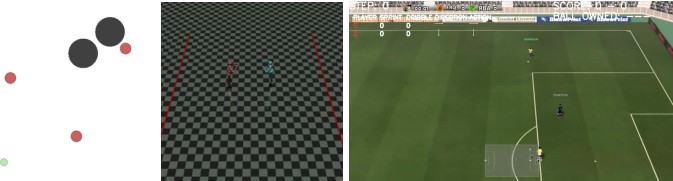

Figure 7: Illustration of experiment environments used in this paper. From left to right: (1) predator prey in MPE, (2) YouShallNotPassHumans in MuJoCo, (3) Google Football.

**Predator-Prey**   There are one good agent, three adversarial agents and two obstacles in the predator-prey environment. The good agent is faster and receive a negative reward for being hit by adversaries while adversaries are slower and are rewarded for hitting the good agent. All agents are initialized randomly in the environment.

**YouShallNotPassHumans**   The YouShallNotPassHumanoids environment [2, 10] creates a two-player competitive game based on MuJoCo, where one humanoid (the runner) is aimed at passing the other humanoid (the blocker) to reach the red line on the opposite side. The environment itself is challenging because it has high dimensional observation space and requires continuous control. In our experiments, we assume that our agent acts as the runner, while the blocker is the target opponent to exploit. We use four independently pre-trained blocker models as opponent targets. For fairness, we generate these targets with exactly four random seeds without any selection. Since the exploitability of different opponent models can vary significantly, results with respect to different opponents are not directly comparable. So we report results with respect to all targets.

**Google Football**   Google Football [21] is a popular and challenging benchmark for MARL. In our experiments, our adaptation policy controls all players in one team, while the opponent target policy controls the other team. Since all players in the same team are fully coopera-tive, we use MAPPO [48] to learn decentralized policy. We conduct experiments in 4 scenar-ios: *academy_3_vs_1_with_keeper* (3vs1, where our agent acts as either defender or attacker), *academy_run_pass_and_shoot_with_keeper* (RPS, defender), and *academy_counterattack_easy* (de-fender). We report the winning rates of adaptation policies for 5 independently pre-trained opponent targets. For attacker, winning rate refers to the percentage of episodes that the attacker scores, while for defender, it refers to the percentage of episodes that the defender prevents the attacker from scoring.

### G.2   Hyper-parameters

In our experiments, we use MAPPO [48] as the base RL algorithm to learn policies on both sides (ours and the target's). MAPPO is an extension of single agent PPO to multi-agent reinforcement learning

within the centralized training decentralized evaluation (CTDE) paradigm. It learns a centralized value function conditioning on global state to promote coordination among the agents. We selected MAPPO as the base learning algorithm due to its simplicity and empirical strong performance.

For the BC-First method, we use exactly the same MAPPO (with the same set of hyper-parameters and the same number of training samples) as CSP to train our adaptation policy, while keeping the target's proxy fixed. In order to ensure that the the performances of BC-First method are not encumbered by an under-trained proxy model, we use the same network structure for the proxy model as the real target model, and train enough steps to make sure that behavior cloning has converged. For the Google Football environment, we use the same hyper-parameters as reported in the MAPPO paper [48] for all scenarios. The hyper-parameters are listed in Table 6, 7, and 8.

Table 6: Hyper-parameters for predator-prey in MPE.

| ppo_epoch | 10 | num_mini_batch | 1 | entropy_coef | 0.01 |
|---|---|---|---|---|---|
| use_gae | True | gamma | 0.99 | gae_lambda | 0.95 |
| critic_lr | 7e-4 | lr | 7e-4 | weight_decay | 0 |
| adam_eps | 1e-5 | n_rollout_threads | 128 | ppo_episode_length | 50 |
| data_chunk_length | 10 | steps | 5M | max_grad_norm | 0.5 |
| bc_regularization_coef | 0.003 | bc_batch_size | 8 | network | RNN |

Table 7: Hyper-parameters for MuJoCo.

| ppo_epoch | 4 | num_mini_batch | 1 | entropy_coef | 0.01 |
|---|---|---|---|---|---|
| use_gae | True | gamma | 0.99 | gae_lambda | 0.95 |
| critic_lr | 7e-4 | lr | 7e-4 | weight_decay | 0 |
| adam_eps | 1e-5 | n_rollout_threads | 100 | ppo_episode_length | 200 |
| data_chunk_length | 10 | steps | 40M | max_grad_norm | 0.5 |
| bc_regularization_coef | 0.1 | bc_batch_size | 256 | network | MLP |

Table 8: Hyper-parameters for Google Football.

| ppo_epoch | 15 | num_mini_batch | 2 | entropy_coef | 0.01 |
|---|---|---|---|---|---|
| use_gae | True | gamma | 0.99 | gae_lambda | 0.95 |
| critic_lr | 7e-4 | lr | 7e-4 | weight_decay | 0 |
| adam_eps | 1e-5 | n_rollout_threads | 50 | ppo_episode_length | 200 |
| data_chunk_length | 10 | steps | 25M | max_grad_norm | 0.5 |
| bc_regularization_coef | 5.0 | bc_batch_size | 256 | network | RNN |

### G.3 Computing Resources

Each seed is run on a GPU server with one NVIDIA P100 GPU, and Intel(R) Xeon(R) Gold 6145 CPU @ 2.00GHz CPU. Each run can finish within 24 hours.

### G.4 Target Models & Dataset Collection

For all the environments, we user different runs of self-play to get the group of targets. In order to ensure a fair comparison, we use the exactly the same number of random seeds to generate these targets, **without any selection**. In our experiments, we observe that different runs with different seeds can produce diverse targets. For instance, in Google Football, different targets may have different tendencies to pick which side to start a attack (left or right). Moreover, as can be seen from Table 4, the difficulties to exploit these targets are diverse. For MuJoCo and Google Football, we use dataset consisting of 5 trajectories. We collect these trajectories using the target model together with a rollout policy. We observe that the 3 kinds of rollout policies: (1) random policy, (2) environment's bot, (3) target itself, do not have significant impacts on the experiment results in the MuJoCo and Google Football environments.

### G.5 Visualization

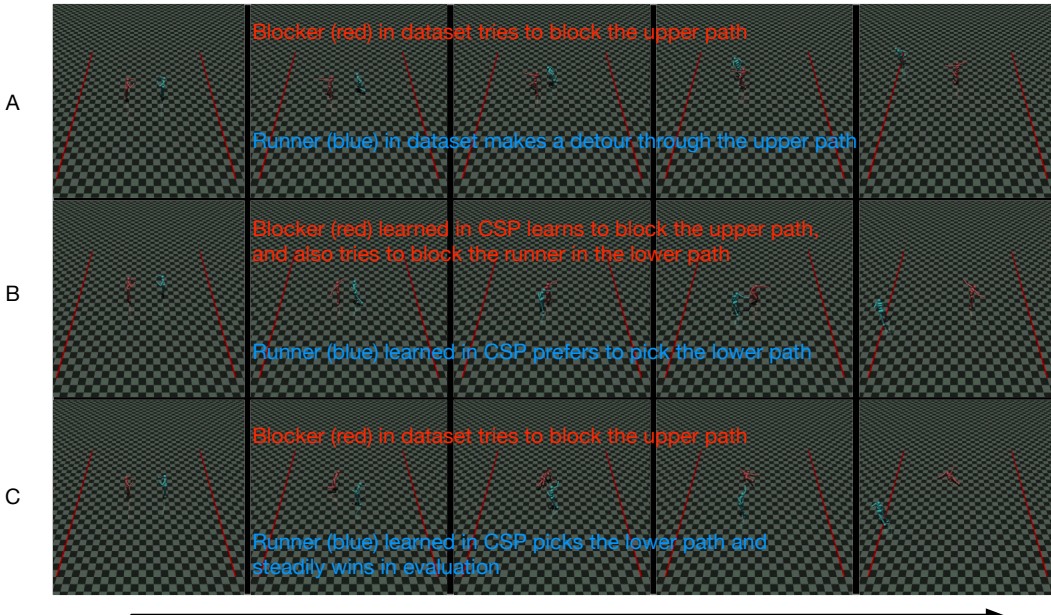

Figure 8: The visualization of policy behavior in MuJoCo. **Top.** The trajectory in dataset, where the blocker is the real opponent policy $\pi_B$ and the runner is the policy which collects the dataset. **Middle.** The trajectory during the training of CSP. Both agents are controlled by CSP. **Bottom.** The evaluation of CSP's runner with the real opponent blocker.

In Figure 8, we further consolidate our claim by visualizing the policy behavior in MuJoCo. As can be seen in this example, in a trajectory contained in dataset, the runner (blue agent) typically makes a detour through the upper path to avoid being pushed down by the blocker (red agent). The opponent blocker also has the tendency to walk upwards to stop the runner. The runner trained by CSP learns to exploit the policy represented by the dataset, and prefers making a detour through the lower path, as can be seen from the middle and bottom lines of replays. Although the behavior of runner going downwards is not contained in the dataset, the blocker trained by CSP can still learns to stop such runner, which makes the runner even more robust and stronger than the real target model. Therefore, when confronted with the real opponent blocker, who only knows how to defend the upper path, the runner trained by CSP steadily wins the game.

### G.6 Reward.

For predator-prey, we use the environment's original reward. For MuJoCo, we use dense rewards for locomotion learning as in previous works [2, 10]. In Google Football, we use both the sparse scoring reward as well as the dense checkpoint reward which awards the attacker according to the ball handler's distance to the goal. Although the rewards for the attacker are provided by the environment, Google Football does not provide rewards for the defender team in academy scenarios. In our experiments, we just use the negative of the attacker's reward to train defender's policy to make the game zero-sum.

### G.7 Explanation of Training & Testing Performances

The observed lower training performance of CSP than the baseline, as shown in Figure 1, is expected behavior since the opponent model used by CSP during training differs from that of BC-First. In BC-First, the opponent model is solely pre-trained on a dataset and remains fixed, leading to significant vulnerabilities in out-of-distribution states which can be easily exploited. Conversely, in CSP, the opponent model is trained in an adversarial manner simultaneously with the exploitation policy. Due to its evolutionary nature, it can compensate for its vulnerabilities and becomes significantly more

challenging to exploit. Consequently, the training performance of BC-First is higher. During testing, both exploitation policies undergo evaluation using the same real target model.

