# OpenReview forum: "Conservative Offline Policy Adaptation in Multi-Agent Games"
_NeurIPS.cc/2023/Conference — NeurIPS 2023 poster_

### Official Review · Reviewer_x4rE · 2023-06-28

**Soundness:** 4 excellent
**Presentation:** 3 good
**Contribution:** 4 excellent
**Rating:** 7
**Confidence:** 4

**Summary:**

The paper introduces a new and previously unconsidered problem in multi-agent reinforcement learning: Offline policy adaptation to a target policy. The paper assumes that a dataset of the target policy is available, however the policy itself is not. This means that the training policy can benefit from analysing the dataset in order to exploit potential weaknesses of the opponent which it will encounter after training. In contrast to the classical offline RL problem, the environment is assumed to be available, just the opponent policy is not. The proposed algorithmic solution maximises policy return against a trained target policy that needs to adhere to the true target policy on in-dataset states (via behaviour cloning), but can perform arbitrary actions elsewhere and is otherwise optimised to minimise performance of the adaptation policy.

**Strengths:**

The proposed problem appears to not have been addressed before even though it seems like a very natural situation and immediately .. potential for improvement over prior approaches. The derived solution of exploiting the target policy only in-data and assuming the worst everywhere else appears straightforward, as it is robust towards different true target policies, yet at the same time realises the potential for exploitation on in-dataset states. The derivation and theoretical considerations are clearly formulated and provide a good foundation for the experimental part. I also found the maze example (section 5) extremely helpful for understanding the overall concept in a limited environment. The experimental evaluation considers a variety of different environments and baselines and shows convincing evidence that the proposed algorithm achieves new state of the art performance.

**Weaknesses:**

In the example environment in section 5, it is made very clear how the adaptation policy can exploit the suboptimal behaviour of the target policy (and which catastrophic mistakes you could make). While the same level of detail is maybe not necessary (or possible) for the other environments, I would have enjoyed a more qualitative comparison: In which way is the target policy here acting sub optimally and in which way is the adaptation policy taking advantage - I realise space is limited but maybe you could showcase 1 or 2 situations in the other environments.

A few details on MAPPO would be good since your whole algorithm builds on it

It is unclear how the behaviour cloning coefficient is chosen and which value(s) are used

I think there is a small mistake in Theorem 4.7, line 203 (one of the mu should be an alpha)

I am not super sure about significance / practical applicability outside of video games - the examples are very illustrative, but I wonder whether an actual problem can be solved with it.

**Questions:**

How is the behaviour cloning coefficient chosen? Is it tuned? Is it sensitive? Do you observe issues in the sense that the trained target policy is not able to achieve both objectives at once (cloning on in-data & minimising adaptation return on out-data)?

Could you provide a practical example where the algorithm provides an advantage? I wonder where in reality an automated agent has to compete against another one that is unknown but data is available

**Limitations:**

One of the most immediate limitations is that opponents are likely not static but will adapt at runtime when faced with an exploiting policy - this is discussed in the conclusion. Potential negative societal impacts are not discussed.

---

> ### Author Rebuttal · Authors · 2023-08-09
>
> We sincerely appreciate the reviewer for recognizing the value and novelty of our work in addressing offline adaptation. We are also pleased that the reviewer finds our theoretical analyses clear and sound and our empirical results significant.
>
> **Q1. In which way is the target policy acting sub-optimally and in which way is the adaptation policy taking advantage in other environments other than the maze example?**
>
> **A1.** In Appendix G.5, we present a visualization in MuJoCo. In this scenario, the runner can take a detour either through the upper path or the lower path to bypass the blocker. The target policy (blocker, red) exhibits a bias toward blocking the upper path, which is captured by the dataset. The adaptation policy (runner, blue) learned by CSP discovers this weakness and learns to take the lower path to steadily win the game.
>
> **Q2. Provide a few details on MAPPO.**
>
> **A2.** We thank the reviewer for bringing up this request. We will add more details in the next revision. MAPPO [1] is an extension of single-agent PPO to multi-agent RL within the centralized training decentralized evaluation paradigm. It learns a centralized value function conditioning on global state to promote coordination among the agents. See Appendix G.2 for hyperparameters.
>
> **Q3. How is $C_\textrm{BC}$ chosen? Is it tuned? Is it sensitive? Do you observe issues in the sense that the trained target policy is not able to achieve both objectives at once?**
>
> **A3.** For each environment, we tune $C_\textrm{BC}$ on one scenario against one opponent. The same hyperparameter is then used for all subsequent experiments in that environment. Please refer to Appendix G for detailed hyperparameters.
>
> 1. *Maze.* The table below shows the test rewards obtained by CSP.
>
>    | $C_\textrm{BC}$ | 0.001 | 0.01 | 0.1  | 1    | 5    | 10   | 100  |
>    | --------------- | ----- | ---- | ---- | ---- | ---- | ---- | ---- |
>    | CASE 1          | 1     | 1    | 1    | 1    | 1    | 1    | 1    |
>    | CASE 2          | 1     | 1    | 1    | 16   | 16   | 16   | 16   |
>
>    In case 1, CSP always finds the optimal solution 1. In case 2, CSP finds the optimal when $C_\textrm{BC}\geq 1$, and shows the same behavior as Self-Play otherwise.
>
> 2. *Predator-Prey in MPE.* We experiment with $C_\textrm{BC}\in$ {0.001, 0.003, 0.01, 0.1, 1, 5, 10} with dataset 10R where the adaptation agent controls adversaries.
>
>    | $C_\textrm{BC}$  | Test Return  |
>    | ---------------- | ------------ |
>    | 0.001            | 33.8 ± 1.3 |
>    | 0.003 (selected) | 34.6 ± 1.6 |
>    | 0.01             | 33.6 ± 2.7 |
>    | 0.1              | 32.3 ± 0.8 |
>    | 1                | 31.3 ± 1.5 |
>    | 5                | 26.8 ± 2.5 |
>
>    Test return for Self-Play: 29.7 ± 2.8, BC-First: 21.3 ± 0.8.
>
> 3. MuJoCo. We experiment with $C_\textrm{BC}\in$ {0.01, 0.1, 1, 5, 10} with one opponent.
>
>    | $C_\textrm{BC}$ | Winning Rate  |
>    | --------------- | ------------- |
>    | 0.01            | 0.68 ± 0.13 |
>    | 0.1 (selected)  | 0.72 ± 0.06 |
>    | 1               | 0.68 ± 0.08 |
>    | 5               | 0.71 ± 0.02 |
>    | 10              | 0.52 ± 0.23 |
>
>    Wining rate for Self-Play: 0.54 ± 0.2, BC-First: 0.65 ± 0.13
>
> 4. *Google Football.* We experiment with $C_\textrm{BC}\in$ {0.01,0.1,1,5,10} with one opponent in 3vs1 attacker scenario.
>
>    | $C_\textrm{BC}$ | Winning Rate  |
>    | --------------- | ------------- |
>    | 0.01            | 0.83 ± 0.04 |
>    | 0.1             | 0.78 ± 0.13 |
>    | 1               | 0.80 ± 0.07 |
>    | 5 (selected)    | 0.88 ± 0.02 |
>    | 10              | 0.88 ± 0.12 |
>
>    Winning rate for Self-Play: 0.76 ± 0.08, BC-First: 0.74 ± 0.21.
>
> We observe that $C_\textrm{BC}$ needs to be tuned for different environments. In the predator-prey, MuJoCo and Google Football environments, even a small coefficient helps in learning to exploit, resulting in better performance of CSP compared to Self-Play. In the predator-prey and MuJoCo, when $C_\textrm{BC}$ is too big, the policy fails to remain conservative and loses to Self-Play or BC-First. Overall, CSP is not very sensitive to $C_\textrm{BC}$, as there exists a wide range of coefficients that allow it to successfully perform conservative adaptation in each environment.
>
> **Q4. Could you provide a practical example outside of video games where the algorithm provides an advantage? I wonder where in reality an automated agent has to compete against another one that is unknown but data is available.**
>
> **A4.** One potential application is in the domain of automated trading agents and online auction for advertisements. In these tasks, multiple automated traders or bidders compete for a variety of resources, aiming to maximize their utility within a limited budget. For example, in advertising, the bidders should select the audience and allocate the budget on each timestep. In such cases, the true policies of other competitors are not accessible.
>
> Moreover, the algorithm can potentially be applied in the field of security. One example is in the prevention for poaching of wild animals, where an automated agent has to decide where to send security guards based on observed trails of poachers. While the dataset of attacker's past behavior is available, the attacker's real policy remains unknown. Therefore, the agent should learn to withstand attacks witnessed in the dataset and also remain cautious when confronted with unfamiliar cases.
>
> **Q5. Are there any potential negative societal impacts?**
>
> **A5.** We have discussed the potential negative impacts in Appendix F. For example, inappropriate use of exploitation algorithms may result in potential exacerbation of inequality in society. However, our algorithm is general-purpose and depends on pre-collected data. We emphasize the importance of stringent laws and regulations that safeguard user privacy to mitigate any adverse effects.
>
> [1] Yu et al. The Surprising Effectiveness of MAPPO in Cooperative Multi-Agent Games, 2021.

---

> > ### Comment · Reviewer_x4rE · 2023-08-17
> > **Rebuttal Response**
> >
> > I thank the authors for their detailed response, which has helped to better understand their work.
> >
> > I have one further question regarding the $C_{BC}$ parameter, the following statement confuses me:
> >
> > > For each environment, we tune $C_{BC}$ on one scenario against one opponent
> >
> > how do you tune it? I.e. do you check for best BC result in terms of log likelihood on the dataset or do you check which $C_{BC}$ value obtains the best winning rate when evaluated online against the real opponent? Because the latter one wouldn't really be offline any more & would use information that is not actually available in the setting that you are considering, right?

---

> > > ### Author Response · Authors · 2023-08-17
> > >
> > > We appreciate the reviewer for bringing attention to the need for a more detailed clarification of the hyperparameter tuning procedure.
> > >
> > > 1. In each environment, all methods are evaluated against a *testing set* of unknown opponent policy and scenario pairs. In order to tune hyperparameter $C_\textrm{BC}$, we assume the existence of one known opponent as a *validation set* of size one (can be any policy, not necessarily to be in the testing set), and tune $C_\textrm{BC}$ based on the online evaluation performance. We then use the same hyperparameter for all test cases in the testing set in that environment without further tuning.
> > >
> > >    Since the lack of explicit labeling for the results in the validation set in the paper might cause confusion to the readers, we will clearly mark the results in the validation set in the next revision. Specifically, the case of 10R adversary in Predator-Prey, the case of opponent No.1 in MuJoCo, and the case of opponent No.2 with 3vs1 attacker scenario in Google football are used for validation. All other test cases are in the testing set, whose opponent policies are unknown. Our algorithm CSP significantly outperforms the baselines on both the validation and testing sets.
> > >
> > > 2. Tuning $C_\textrm{BC}$ against one opponent is not overly restrictive in many real-world applications. Although the real opponent's policy is unavailable, obtaining an alternate strategy in that environment for tuning $C_\textrm{BC}$ can often be feasible. For example, we can utilize publicly available policies online or employ a rule-based agent.
> > >
> > > 3. We believe that automatically determining $C_\textrm{BC}$ without the need for hyperparameter tuning is an interesting research problem. As discussed in the Conclusion section (Lines 347-349), one approach is to use our algorithm to train a population of risk-free adaptation policies with different values of $C_\textrm{BC}$ and subsequently employ bandit algorithms to determine the optimal choice for online evaluation.

---

> > > > ### Comment · Reviewer_x4rE · 2023-08-18
> > > > **Rebuttal Response 2**
> > > >
> > > > Thank you for your insightful comments on the hyper parameter tuning as well as addressing all my other concerns. I share your view presented in (2), so I will raise my score to a 7.

---

> > > > > ### Author Response · Authors · 2023-08-18
> > > > > **Thanks for raising the score to 7!**
> > > > >
> > > > > We express our gratitude to the reviewer for raising the score to 7! We sincerely appreciate the reviewer's perceptive feedback and suggestions, which significantly contributed to enhancing the quality of our work.

---

### Official Review · Reviewer_bBXB · 2023-07-05

**Soundness:** 3 good
**Presentation:** 3 good
**Contribution:** 3 good
**Rating:** 6
**Confidence:** 4

**Summary:**

This paper considers multi-agent games where you have access to the
environment and you have a previously collected dataset from the
opponent you will be facing.  So it is offline in the sense that you
only have data on the opponent, but online in the sense that you can
practice as much as you want in the environment -- you just need to
supply your own opponent policy, possibly informed by the opponent
dataset you are given.  The paper proposes an RL algorithm (called
Constrained Self Play) whose loss function incentivizes adherence to
the dataset for states in the dataset and worst-case optimization for
states not in the dataset.  It provides related theoretical analysis.
CSP is tested empirically on several multi-agent adversarial games.


**Strengths:**

The idea for CSP is interesting and well supported by the arguments in
the narrative.  The problem setup is an interesting one to consider as
many adversarial applications will match it reasonably well.  The
games tested in the empirical study are sufficiently complex to show
non-trivial behavior.


**Weaknesses:**

My score is limited by the fact that I did not find the theoretical
analysis useful.  My concerns with it are as follows:

- The statements that "we prove that CSP learns a near-optimal
  risk-free offline adaptation policy" are overstatements given the
  differences between the algorithm and the analysis.  There are a
  number of factors that mean there *is* risk in applying this method.

- Objective 2 can not be optimized in general.  Objective 5 is designed
  to address some of the issues by replacing the constraint with a KL
  divergence.  However, there is no way to guarantee or measure delta.
  It will be affected by statistical error due to low sample size in
  the dataset as well as failures in the optimization thus leading to
  an unknown and arbitrarily large delta.

- Theorem 4.8 assumes the minimizations in Eq 6 can be performed.  The
  actual CSP algorithm has no guarantees of those minimums being
  reached.

- Since the theory provides no true guarantees about CSP, we can only
  hope it provided some insights that led to its development.  I believe
  the CSP idea is interesting and can be explained without the theory.

- I would rather see the space that was used for theoretical analysis
  be used to more completely explain the algorithm and the design
  tradeoffs.  Also, that space could be used to pull in more of the
  empirical results.  As it is, competitors that are designed just to
  do imitation or self-play would not be expected to perform well at
  this task and some of the wins don't look very big.  It would be good
  to see them illustrated more thoroughly.


**Questions:**

Some smaller comments:

line 60: "improve in training and testing".  I'm not sure what the
training part of this refers to.  In figure 1, the training
performance of CSP is below the baseline.

---

> ### Author Rebuttal · Authors · 2023-08-09
>
> We thank the reviewer for acknowledging the significance of the offline policy adaptation problem and for recognizing that our algorithm CSP is "well supported", and our empirical study demonstrate "non-trivial behavior". Your positive feedback is greatly appreciated.
>
> **Q1. The statements that "we prove that CSP learns a near-optimal risk-free offline adaptation policy" are overstatements due to two concerns: (1) the KL divergence $\delta$ can be hard to minimize in practice by statistical error due to low sample size in the dataset as well as failures in the optimization, (2) self-play is not guaranteed to converge.**
>
> **A1.** We appreciate the reviewer for bringing up these points. We will refine this statement to be more accurate in the next revision. The revised statement is "if CSP converges, it learns a near-optimal risk-free offline adaptation policy given that the target agent's policy within the support of the dataset can be well approximated". We will provide clarification to the reviewer's two concerns as follows.
>
> 1. The inclusion of $\delta$ in our analysis is not overly restrictive. To clarify, $\delta$ is an upper bound on the KL divergence between the learned proxy model $\mu$ and real target policy $\pi_B$ (Line 205) **within the support of the dataset**: $\max_{s\in D}D_\textrm{KL}(\pi_B(\cdot|s)\|\mu(\cdot|s))\leq \delta$. It quantifies the extent to which an agent model can imitate the real target policy within the dataset, whose success is a necessary condition for any further adaptations. Additionally, as commented by the reviewer, our theoretical analysis through KL divergence $\delta$ is a critical motivation to the development of CSP. Our theoretical findings in Theorem 4.8 reveals a clear connection between $\delta$ and adaptation performance, demonstrating that a smaller approximation gap results in better performance.
>
>    Accurately fitting a deterministic policy within the dataset to minimize $\delta$ is relatively straightforward given sufficient representational capacity and learning steps. However, we acknowledge that controlling $\delta$ becomes more challenging for highly stochastic policies with limited data.
>
> 2. We concur with the reviewer's observation that, although the converged solution of CSP possesses the theoretical properties presented in Theorem 4.8, the convergence of CSP itself is not guaranteed. The development of provable self-play algorithms for zero-sum games [1] remains an active area of research in RL theory. Nevertheless, empirical evidence from RL methods utilizing simple self-play [2, 3, 4] has shown that they can train max-min policies with strong performance in video games. Given that our paper primarily focuses on addressing the unique challenges posed by offline adaptation, we have incorporated a self-play approach in our algorithm for efficiency.
>
> Despite these concerns, the experimental results show CSP's superiority over the baselines. In particular, Figure 3 illustrates that CSP's testing performance, on average, surpasses its training performance (Lines 310-324). These results suggest that the CSP does motivate the adaptation agent to keep conservative with respect to the target's out-of-distributon policy in training, while also enabling it to discover risk-free exploitation opportunities.
>
> **Q2. What does "improve in training and testing" in line 60 mean? Why is the training performance of CSP below the baseline?**
>
> **A2.** We mean that both the training and testing performance of CSP improve as the number of training steps increases in Figure 1. Conversely, for BC-First, the improvement over steps in training does not result in better evaluation performance. It shows that the BC-First baseline quickly overfits the proxy model, suffering from distributional shift.
>
> "The training performance of CSP is below the baseline" is expected behavior, because the opponent model of CSP in training are not the same as BC-First. In BC-First, the opponent model is pre-trained on dataset only and fixed, which can have significant weaknesses on out-of-distribution states, and can be easy to exploit. In CSP, the opponent model is trained adversarially against the exploitation policy simultaneously. Because it is evolving, it can make up its weaknesses and becomes much harder to exploit. Consequently, the training performance os BC-First is higher. In testing, both exploitation policies are evaluated with the same real target model.
>
> **References**
>
> [1] Yu Bai and Chi Jin. Provable Self-Play Algorithms for Competitive Reinforcement Learning. In *Proceedings of the 37th International Conference on Machine Learning*, 2020.
>
> [2] Ye et al. Towards Playing Full MOBA Games with Deep Reinforcement Learning. In *Advances in Neural Information Processing Systems*, 2020.
>
> [3] Ye et al. Mastering Complex Control in MOBA Games with Deep Reinforcement Learning. In *Proceedings of the AAAI Conference on Artificial Intelligence*, 2020.
>
> [4] Zhao et al. AlphaHoldem: High-Performance Artificial Intelligence for Heads-Up No-Limit Poker via End-to-End Reinforcement Learning. In *Proceedings of the AAAI Conference on Artificial Intelligence*, 2022.

---

### Official Review · Reviewer_71Zz · 2023-07-06

**Soundness:** 2 fair
**Presentation:** 3 good
**Contribution:** 2 fair
**Rating:** 6
**Confidence:** 4

**Summary:**

In this paper, the authors target the offline policy adaptation problem in multi-agent games where "offline" indicates no interactive target agents and an accessible environment. The problem is challenging as it inherits the distributional shift challenge from offline RL, and is likely to lead to risky actions. To tackle these problems, the authors introduced the novel conservative offline adaptation objective to optimize the worst-case performance and proposed the constrained self-play, i.e., CSP algorithm to simultaneously train an adaptation policy and a proxy model subject to regularization.

**Strengths:**

1. This paper is well-written and easy to follow with good visualizations.
2. There are plenty of theoretical results supporting the proposed solution.
3. Extensive experiments are done to validate the proposed solution.

**Weaknesses:**

1. Novelty. The proposed offline policy adaptation problem seems to be a special case of opponent modeling problem [1,2] where the static opponent's data are available. Therefore, the studied problem is not new.

2. Theoretical. I did not see the connection of Theorem 4.7 with the CPS algorithm.

[1] Nashed, Samer, and Shlomo Zilberstein. "A survey of opponent modeling in adversarial domains." Journal of Artificial Intelligence Research.

[2] Haobo Fu, et al. "Greedy when Sure and Conservative when Uncertain about the Opponents." ICML, 2022.

**Questions:**

1. If in the cooperative games, the objective for CSP remains the same min-max problem?
2. What is the connection between Theorem 4.7 with the CPS algorithm?

**Limitations:**

The authors leave the limitation where not all agents are rational in future work.

---

> ### Author Rebuttal · Authors · 2023-08-09
>
> **Q1. The proposed offline policy adaptation problem seems to be a special case of opponent modeling problem [1,2] where the static opponent's data are available.**
>
> **A1.** We have discussed the differences of our work from previous works in opponent modeling in the Introduction (Lines 39-47) and the Related Work (Lines 91-97). Opponent modeling (sometimes called opponent exploitation in non-cooperative settings) is a broad and extensively studied area [1]. The offline adaptation problem we studied in this work is a largely unstudied yet significant and challenging problem within the broader scope of opponent modeling. Different from previous works that assume access to the opponent's policy in training [2, 3] and from others that assume no prior knowledge [4, 5], we leverage the opponent's dataset and maintain access to the environment to train an exploitation policy.
>
> Our paper presents theoretical and empirical evidence that previous methods, lacking a conservative perspective on the out-of-distribution (OOD) behavior of the opponent, are insufficient in effectively addressing the problem and suffer from considerable performance degradation. In contrast, our algorithm CSP performs worst-case optimization for OOD policy. We prove that its converged solution is close to optimal, with the gap bounded by the approximation accuracy of the opponent's policy on states in the dataset. Experimental results demonstrate that CSP outperforms the baselines.
>
> The second paper mentioned by the reviewer, GSCU [4], has been discussed in Lines 95-97 of our paper. However, it actually focuses on zero-shot exploitation of completely unknown opponents, which is different from our setting. In the table below, we show the experiment results of GSCU on the predator-prey task where we control the prey agent. We can observe that CSP significantly outperforms the baselines. GSCU fails to effectively exploit the opponent due to its lack of utilization of prior information about the opponent.
>
> | Dataset       | 10R           | 100R          | 100S          |
> | ------------- | ------------- | ------------- | ------------- |
> | **CSP(Ours)** | **-16.7±0.4** | **-17.5±0.4** | **-17.1±0.1** |
> | BC-First      | -29.8±4.0     | -21.8±1.6     | -18.5±0.4     |
> | GSCU          | -27.7±2.8     | -27.7±2.8     | -27.7±2.8     |
> | Self-Play     | -17.6±1.5     | -17.6±1.5     | -17.6±1.5     |
> | Oracle        | -16.0±0.3     | -16.0±0.3     | -16.0±0.3     |
>
> *: We directly use the trained weights of GSCU model provided by the original paper to test in our experiment settings. Since GSCU performs Bayesian updates in testing, we independently evaluate GSCU with three random seeds and report the mean and standard error. GSCU does not depend on the dataset. Exploiting policies control good agents in this task.
>
> **Q2. Explain the connection of Theorem 4.7 with the CSP algorithm.**
>
> **A2.** Theorem 4.7 bridges the gap between Objective 2 (Line 172) and Objective 5 (205), and serves as a necessary prerequisite to establish our main result, Theorem 4.8, which proves the proximity of the converged solution obtained by the CSP to optimality.
>
> While Objective 2 (Line 172) produces an optimal risk-free adaptation policy, satisfying its dataset consistency constraint is difficult. To address this issue, we replace the constraint with a KL divergence and propose Objective 5 (Line 205): $\max_\pi\min_\mu J(\pi, \mu)\ s.t.\ \max_{s\in D}D_{KL}(\pi_B(\cdot|s)\|\mu(\cdot|s))\leq \delta$. This modified objective becomes feasible for learning in CSP. What remains is to establish a connection between the solutions obtained by optimizing these two objectives. Theorem 4.7 demonstrates that the gap in return can be bounded by the KL divergence of two policies of the target agent. Building upon this result, Theorem 4.8 concludes that the solution obtained by CSP (Objective 5) is also in proximity to the optimal solution (Objective 2), with the gap bounded by the KL divergence between the learned proxy model and the real target policy on the dataset.
>
> **Q3. Why does the objective for CSP remains the same min-max problem in cooperative games?**
>
> **A3.** The objective of CSP in cooperative games maintains the same min-max structure because it promotes adherence to the policy exposed by dataset for states within the dataset, while aiming to optimize for worst-case performance for states not in the dataset (Objective 5, Line 205). Therefore, in cooperative games, the policy regularization term incentivizes the adaptation agent to collaborate with the target agent on states within the dataset, and the minimization over opponent proxy $\mu$ encourages the adaptation agent to be conservative and not to rely on the target agent's OOD policy.
>
> We have explained in more details about the implementation of CSP in cooperative games in Appendix E. CSP optimizes the worst-case return with respect to the adaptation agent's utility with policy regularization. No matther whether the game is competitive or cooperative, the minimization over $\mu$ in training can be achieved by setting the target agent's reward function to be the negative of our adaptation agent's reward function. We then train $\mu$ to maximize this reward with policy regularization in self-play.
>
> **References**
>
> [1] Albrecht et al. Autonomous agents modelling other agents: A comprehensive survey and open problems, 2018.
>
> [2] Gleave et al. Adversarial Policies: Attacking Deep Reinforcement Learning, 2020.
>
> [3] Johanson et al. Computing robust counter-strategies, 2007.
>
> [4] Fu, et al. Greedy when Sure and Conservative when Uncertain about the Opponents, 2022.
>
> [5] Lupu et al. Trajectory Diversity for Zero-Shot Coordination, 2021.

---

> > ### Comment · Reviewer_71Zz · 2023-08-19
> > **Thank the authors for the detailed explanation**
> >
> > Thank the authors for the detailed explanation which resolved most of my concerns.

---

> > > ### Author Response · Authors · 2023-08-19
> > > **Thanks for raising the score to 6!**
> > >
> > > We are grateful to the reviewer for raising the score to 6! We sincerely appreciate the reviewer's insightful comments, which have greatly improved the clarity and quality of our paper.

---

### Official Review · Reviewer_dBj1 · 2023-07-06

**Soundness:** 3 good
**Presentation:** 3 good
**Contribution:** 3 good
**Rating:** 6
**Confidence:** 3

**Summary:**

This paper presents a novel approach to offline policy adaptation in multi-agent games, focusing on leveraging the target agent's behavior dataset and access to the game environment during training. The authors identify the challenges of distributional shift and risk-free deviation in offline policy adaptation and propose a new learning objective called conservative offline adaptation to address these issues. They introduce an efficient algorithm, Constrained Self-Play (CSP), which trains an adaptation policy and a proxy model with regularizations. The paper provides theoretical proof of CSP's effectiveness and demonstrates its superiority over non-conservative methods through empirical evaluations in various environments, including Maze, predator-prey, MuJoCo, and Google Football.



**Strengths:**

1. The specific problem of offline policy adaptation is a highly meaningful and novel issue, particularly with a wide range of potential applications in the industrial sector.

2. The article is easy to understand, with a comprehensive background and related work section.

3. Although the ideas presented are simple and heuristic, they are supported by theoretical proofs and significant experimental results.




**Weaknesses:**

1. The paper does not discuss the additional computational and time costs associated with offline adaptation.

2. Offline adaptation and opponent modeling share some similarities, so it is important to discuss their differences and include comparative experiments to highlight the distinctions between the two approaches, rather than just comparing self-play and behavior cloning methods.

3. The term "risk-free" is mentioned several times in the paper. Please provide a more specific description of what "risk" refers to in the context of the paper, or provide some examples to clarify its meaning.

4. The paper discusses issues related to human-AI interaction, and it would be beneficial to address the associated ethical concerns in the context of the proposed approach.




**Questions:**

Please refer to Weaknesses Section



**Limitations:**

Please refer to Weaknesses Section

---

> ### Author Rebuttal · Authors · 2023-08-09
>
> We express our gratitude to the reviewer for commenting that offline policy adaptation is "highly meaningful and novel", and the theoretical proofs and experimental results effectively support our idea.
>
> **Q1. Are there any additional computational and time costs associated with offline adaptation?**
>
> **A1.** The computational and time efficiency of our algorithm CSP is comparable to that of Self-Play. While CSP incorporates an additional policy regularization step, it mainly entails (1) sampling a batch of data from a pre-collected dataset, and (2) minimizing the regularization loss through backpropagation (refer to lines 11 and 13 of Algorithm 1 in Appendix C). The incurred computational and time costs are marginal compared to other RL procedures.
>
> It's worth mentioning that both CSP and Self-Play use half of the environment steps to train the exploitation policy and the other half to train the opponent model due to alternative update. In contrast, BC-First and other opponent exploitation methods which rely on a fixed pre-trained opponent model can use all steps to train the exploitation policy. In our experiments, we ensured that all methods were given the same overall number of environment steps. CSP outperforms BC-First, even when utilizing only half of the data for training the exploitation policy.
>
> The table below shows the average training time of different methods in Google Football with 25M overall environment steps.
>
> | Method        | CSP (Ours) | Self-Play | BC-First  |
> | ------------- | ---------- | --------- | --------- |
> | Training Time | 12h 17min  | 12h 10min | 10h 55min |
>
> **Q2. Offline adaptation and opponent modeling share some similarities, so it is important to discuss their differences and include comparative experiments to highlight the distinctions between the two approaches.**
>
> **A2.** We have discussed the differences of our work from previous works in opponent modeling in the Introduction (Lines 39-47) and the Related Work (Lines 91-97). We will refine the discussion in the next revision.
>
> Opponent modeling (sometimes called opponent exploitation) is a broad and extensively studied area [1]. The offline adaptation problem we studied in this work is a largely unstudied yet significant and challenging problem within the broader scope of opponent modeling. Some previous works assume access to the target agent's policy in training [2, 3], while some others assume no prior knowledge [4, 5] and concentrate on zero-shot exploitation or cooperation. In contrast, as the reviewer commented, our work introduces a novel setting of offline policy adaptation, where we leverage the target agent's dataset and access to the environment to train an adaptation policy.
>
> Our paper reveals both theoretically and empirically that methods failing to adopt a conservative perspective on the opponent's out-of-distribution (OOD) policy are inadequate in effectively addressing the problem. Our CSP algorithm promotes worst-case optimization for OOD policy and significantly outperforms the baselines. In addition to Self-Play and BC-First, we also compare CSP with RNR [3], an exploitation algorithm that minimizes the distance to NE during exploitation, in Section 6.3. Notably, CSP outperforms RNR as well.
>
> Finally, it is crucial to note that our paper is orthogonal to other opponent modeling subareas, including designing efficient models for predicting opponent behavior [6, 7], and addressing non-stationary opponents [4], which we have discussed in the Conclusion (Lines 342-349).
>
> **Q3. Provide a more specific description of what "risk" refers to in the context of the paper.**
>
> **A3.** We appreciate the reviewer for pointing it out. In the context of this paper, "risk" intuitively refers to the likelihood of the adaptation policy experiencing a decline in performance in evaluation. Formally, let $\pi_B$ denote the unknown real target policy, $\pi^*$ represent the adaptation policy, and $\mu$ be any proxy model utilized in training $\pi^*$ as a substitute for $\pi_B$. Consider $J(\pi^*, \mu)$ as the performance of $\pi^*$ in training and $J(\pi^*, \pi_B)$ as its testing performance against the real target agent. Risk refers to chances that $J(\pi^*, \pi_B)< J(\pi^*, \mu)$, implying that the performance observed during training may not be guaranteed in evaluation.
>
> For examples clarifying the concept, please refer to the right part of Figure 1 and the left part of Table 1. Non-conservative adaptation methods may make erroneous generalization. The evalution performance is much lower than that in training.
>
> On a contrasting note, our paper proves (Lines 190-192) that conservative offline adaptation (COA) is risk-free: $J(\pi^*, \pi_B)-J(\pi^*, \mu)\geq0$. Additionally, in section 6.2, we also empirically demonstrate that the average gap for our algorithm CSP is positive, but the baseline method remains highly risky.
>
> **Q4. Explain the associated ethical concerns since the paper discusses issues related to human-AI interaction.**
>
> **A4.** We have discussed the potential negative social impacts in Appendix F. For example, companies with access to extensive data may exploit their customers more effectively. However, our algorithm is general-purpose and depends on pre-collected data. We emphasize the importance of stringent laws and regulations that safeguard user privacy to mitigate any adverse effects.
>
> **References**
>
> [1] Albrecht et al. Autonomous agents modelling other agents: A comprehensive survey and open problems, 2018.
>
> [2] Gleave et al. Adversarial Policies: Attacking Deep Reinforcement Learning, 2020.
>
> [3] Johanson et al. Computing robust counter-strategies, 2007.
>
> [4] Fu, et al. Greedy when Sure and Conservative when Uncertain about the Opponents, 2022.
>
> [5] Lupu et al. Trajectory Diversity for Zero-Shot Coordination, 2021.
>
> [6] He et al. Opponent modeling in deep reinforcement learning, 2016.
>
> [7] Rabinowitz et al, Machine theory of mind, 2018.

---

> > ### Comment · Reviewer_dBj1 · 2023-08-17
> > **Reply to the authors**
> >
> > Thanks for your detailed reply, I have raised my score.

---

> > > ### Author Response · Authors · 2023-08-17
> > > **Thanks for raising the score!**
> > >
> > > We would like to thank the reviewer for raising the score! We really appreciate the valuable comments and suggestions from the reviewer, which greatly help improve our work.

---

### Comment · Area_Chair_RSTx · 2023-08-18
**Please engage in discussion phase**

Dear Reviewers bBXB and 71Zz,

Please carefully consider the author's rebuttal to your concerns raised. The discussion period is now coming to an end and we need to ensure a fair process.

Many thanks!
AC

---

### Decision · Program_Chairs · 2023-09-21

**Decision:**

Accept (poster)

**Comment:**

This paper introduces a new method for adapting to opponents using offline data.

During the active discussion phase the authors managed to address the overwhelming majority of open problems from the reviewers resulting in a few of the reviewers raising their scores.

As a consequence, this paper now is a unanimous accept and should be published at the conference.